# Rapid lightsheet fluorescence imaging of whole *Drosophila* brains at nanoscale resolution by potassium acrylate-based expansion microscopy

Xuejiao Tian [1,2,3,9], Tzu-Yang Lin [4,9], Po-Ting Lin[1], Min-Ju Tsai[1], Hsin Chen[1], Wen-Jie Chen[5,6], Chia-Ming Lee[1], Chiao-Hui Tu[1], Jui-Cheng Hsu [1], Tung-Han Hsieh[1], Yi-Chung Tung [1], Chien-Kai Wang[7], Suewei Lin [6], Li-An Chu [8], Fan-Gang Tseng [1,2,3], Yi-Ping Hsueh [6], Chi-Hon Lee [4], Peilin Chen [1] & Bi-Chang Chen[1,4] ✉

Taking advantage of the good mechanical strength of expanded *Drosophila* brains and to tackle their relatively large size that can complicate imaging, we apply potassium (poly)acrylate-based hydrogels for expansion microscopy (ExM), resulting in a 40x plus increased resolution of transgenic fluorescent proteins preserved by glutaraldehyde fixation in the nervous system. Large-volume ExM is realized by using an axicon-based Bessel lightsheet microscope, featuring gentle multi-color fluorophore excitation and intrinsic optical sectioning capability, enabling visualization of Tm5a neurites and L3 lamina neurons with photoreceptors in the optic lobe. We also image nanometer-sized dopaminergic neurons across the same intact iteratively expanded *Drosophila* brain, enabling us to measure the 3D expansion ratio. Here we show that at a tile scanning speed of ~1 min/mm³ with $10^{12}$ pixels over 14 hours, we image the centimeter-sized fly brain at an effective resolution comparable to electron microscopy, allowing us to visualize mitochondria within presynaptic compartments and Bruchpilot (Brp) scaffold proteins distributed in the central complex, enabling robust analyses of neurobiological topics.

Imaging at the nanoscale level underlies current efforts to understand the molecules and circuits important in neuroscience. However, the resolving power of conventional light microscopes is limited to a few hundred nanometers, termed the Abbe limit. Overcoming the Abbe limit via a bottom-up chemical approach has been realized by expansion microscopy (ExM), whereby a biological specimen is physically magnified by superabsorbent polymers[1–6]. However, given the expanded specimen volume and resulting decreased fluorophore density inherent in this approach, technologies for rapid high-resolution imaging that minimizes photobleaching are greatly

¹Research Center for Applied Sciences, Academia Sinica, Taipei 11529, Taiwan. ²Nano Science and Technology Program, Taiwan International Graduate Program, Academia Sinica, Taipei 11529, Taiwan. ³Department of Engineering and System Science, National Tsing Hua University, Hsinchu 300, Taiwan. ⁴Institute of Cellular and Organismic Biology, Academia Sinica, Taipei 11529, Taiwan. ⁵Taiwan International Graduate Program in Interdisciplinary Neuroscience, National Cheng Kung University and Academia Sinica, Taipei 11529, Taiwan. ⁶Institute of Molecular Biology, Academia Sinica, Taipei 11529, Taiwan. ⁷Department of Mechanical Engineering, National Taiwan University, Taipei 106319, Taiwan. ⁸Department of Biomedical Engineering and Environmental Sciences, National Tsing Hua University, Hsinchu, Taiwan. ⁹These authors contributed equally: Xuejiao Tian, Tzu-Yang Lin. ✉e-mail: chenb10@gate.sinica.edu.tw

needed, with lightsheet microscopy representing one such methodology[7-9]. Recently, a combination of 4x or 8x ExM with lattice lightsheet microscopy (LLSM) enabled rapid deep-tissue 3D volumetric imaging at sufficient superresolution to resolve synaptic connections across different specimens[10,11]. Due to the orientation in which samples are mounted in ExLLSM, 3D volumetric imaging is achieved by objective scanning (Supplementary Fig. 1a). However, that scanning method may suffer from lightsheet synchronization over long distances in thick samples and may also be limited by the objective z-piezo operational distance[12], which is a more serious concern for samples expanded enormously to gain a higher spatial resolution. Moreover, for protein retention during the expansion process, strong chemical fixation such as by means of glutaraldehyde (GA) treatment may be considered for imaging transgenic fluorophore proteins.

In addition, whole-brain scale mapping of the neural architecture at higher spatial resolution may be further boosted by increasing the sample expansion ratio. By using different hydrogels, expansion factors of 10- to 20-fold can be realized through iterative ExM (iExM) or by X10 microscopy in a single step[3,13-16]. However, these methods have mainly been applied to cultured cells or thinly sliced tissues (see summary in Supplementary Table 1), with any expansion of thick tissue slices (>100 μm) requiring a greater expansion factor in a single step necessitating different gel chemistry. From our experience, the N, N′-dimethylacrylamide (DMAA) and sodium acrylate (SA) used in the original 10X hydrogel recipe are both unstable water-absorbent polymers that yield particularly watery gels, prohibiting further expansions for image acquisition. Moreover, DMAA may undergo self-crosslinking in the absence of an additional crosslinking reagent and it degrades rapidly, rendering monomer cocktail storage difficult under freezing conditions. Thus, the monomers have to be freshly made each time. Furthermore, upon increasing the magnification, the expanded gel becomes watery, so it does not readily maintain its shape during image acquisition and biological structures in the embedded samples are easily damaged (Supplementary Fig. 1b), especially when a large field of view is required, such as for expanded fly brains at ~100-fold their original volume[11]. Importantly, to optimize complex image processing steps such as transformation, deformation, and stitching in expansion lightsheet microscopy, it is necessary to directly record images without oblique sample scanning in cases where a long travel distance through a sample is warranted. To achieve that, the greatly expanded gel must exhibit high mechanical strength[17].

To prepare large samples with sufficient physical strength for imaging in expansion microscopy, we adapted the concept of sol-gel synthesis for inorganic silica aerogels in which tetraethoxysilane or sodium silicate are used as precursors for polymerization[18-20]. Considering how the chosen counter-ion alters gel morphology or physical properties[21], we used potassium acrylate (KA) or potassium polyacrylate (PKA) as monomer or oligomer, respectively, for hydrogel formation (see summary in Supplementary Table 1)[22]. The resulting bulky hydrogels exhibited optimal strength for lightsheet-linked ExM. We also built an axicon-based Bessel beam lightsheet microscope (ΔBLX, where Δ symbolizes the conical surface of an axicon lens) with two long working distance objectives to image centimeter-sized expanded fly brains, allowing us to genetically target mitochondria inside the neural circuit and immunostain with Brp[Nc82] to establish nanoscopic organization of Bruchpilot scaffold proteins.

## Results

### KA-ExM and re-KA-ExM on whole fly brain for imaging by Bessel lightsheet microscopy

To prepare expanded samples with good mechanical strength and effective preservation of fluorophores, we have developed a simple protocol for in situ synthesis of KA hydrogels compatible with GA-fixed specimens, using potassium hydroxide and acrylic acid to replace the sodium acrylate in the traditional expansion anchoring step for

reaction with the cross-linker, N, N′-methylenebisacrylamide (MBA). Potassium persulfate acts as the initiator and N, N, N′, N′-tetra-methylethylenediamine acts as the gelation accelerator, as illustrated in Fig. 1a and described in the "Methods". Note that DMAA is not used in our in situ KA-ExM protocol and the acrylic polymer is newly synthesized, resulting in a well-controlled gelation process. Moreover, we obtained a more rigid gel by using $K^+$ instead of $Na^+$ as the counter-ion for polymerization of the acrylic monomers (Supplementary Fig. 1c, d). We measured the achievable expansion factor by imaging the stained nuclei of cultured COS-7 cells. Our results (Supplementary Fig. 1e, f) revealed an isotropic expansion factor of ~7x based on the measurement of identical cells pre- and post-expansion (Supplementary Fig. 1g).

To test our protocol for large samples, we applied in situ KA-based ExM to the *Drosophila melanogaster* brain (Fig. 1b, showing a freestanding DAPI-stained fly brain under an epi-fluorescence microscope using 405 nm laser excitation), resulting in initial expansion from ~540 μm to ~5 mm in length. The respective detailed protocol is presented in the "Methods". Since this in situ KA-based hydrogel displayed good mechanical strength and was even suitable for sectioning into thin slices using a vibratome (Supplementary Movie 1), we could easily mount it as a freestanding hydrogel to perform direct sample scanning by lightsheet microscopy without gel deformation. In order to image the entire expanded fly brain, we built a ΔBLX equipped with two long working distance (WD) objectives: a customized excitation objective (numerical aperture (N.A.) = 0.5, WD = 11.7 mm) to provide good optical sectioning capability, and an Olympus detection objective (N.A. = 0.6, WD = 8 mm) to accommodate the thickness of the freestanding expanded gel (Fig. 1c and "Methods")[23]. Direct sample scanning was performed using a voice coil stage (V308, Physik Instrumente) featuring a long travel distance of $Z = 7$ mm, which was run in closed loop and analog input mode, allowing us to tile scan the freestanding hydrogel that was glued onto an L-shaped sample holder (Fig. 1c and Supplementary Fig. 1d). The unit volume comprised ~1400 x 2048 pixels x Z steps at a pixel size of 0.325 μm, corresponding to 457 μm x 665 μm x Z mm (x, y, z), where $Z \leq 7$ depending on gel thickness, with 20% overlapped region applied for stitching (Fig. 1d). A detailed optical scheme for our ΔBLX system and additional parameters is presented in Supplementary Fig. 2 and Supplementary Table 2.

### Visualization of 3D whole expanded *TH-GLA4*-expressing fly brain by ΔBLX multi-color imaging

The rate-determining reaction in the biosynthetic pathway of *Drosophila melanogaster* that produces the neurotransmitter dopamine is mediated by tyrosine hydroxylase (TH)[24]. To map the distribution and wiring patterns of all *TH-Gal4* neurons in the adult fly brain, we used *TH-GAL4*, 20XUAS-6XGFP/+ *Drosophila melanogaster* as an example for in situ KA-ExM whole fly brain lightsheet imaging of transgenic fluorophore proteins. To provide a longer imaging duration and less photobleaching for red wavelength-optimized lightsheet microscopy, we used an anti-GFP antibody boosted with Streptavidin Alexa-635 (SA-635) dye before sample expansion and determined the 3D expression pattern of the transgene *TH-GAL4* in the expanded fly brain. In the upper left of Fig. 1e, we present a snapshot grayscale image of an original fly brain (~600 μm in length) under epi-fluorescence microscopy. Following in situ KA-ExM treatment, this fly brain expanded to 4.6 mm, representing an ~8x expansion factor. Two-color maximum intensity projection (MIP) images for DAPI-stained nuclei (purple) and GFP-labeled dopaminergic neurons (heatmap) in the 8x and 13x (see below) fly brain are shown in the middle and bottom of Fig. 1e. Volumetric imaging was performed by means of ΔBLX using 5×12 tiles, taken ~2 hrs at a voxel resolution of ~0.325 x 0.325 x 3 μm³, which is equivalent to a resolving volume of ~40 x 40 x 325 nm³ at the ~8x expansion ratio. The 3D renderings for this two-color 8x expanded *TH-GAL4* fly brain are presented in Supplementary Movie 2.

Following iterative expansion, the same fly brain specimen was further expanded to ~13x, hereafter termed *re*-KA-ExM. Again, ΔBLX imaged this iteratively expanded fly brain (~7.5 mm) in 8x20 tiles (represented by the red box in the bottom panel of Fig. 1e), captured

over ~12 hrs at an equivalent voxel resolution of ~25 x 25 x 230 nm$^3$ given the ~13x expansion ratio. We performed a detailed 3D comparison at high spatial resolution for the in situ KA-ExM (8x) and *re*-KA-ExM (13x) data (Supplementary Fig. 3 and Supplementary Table 3).

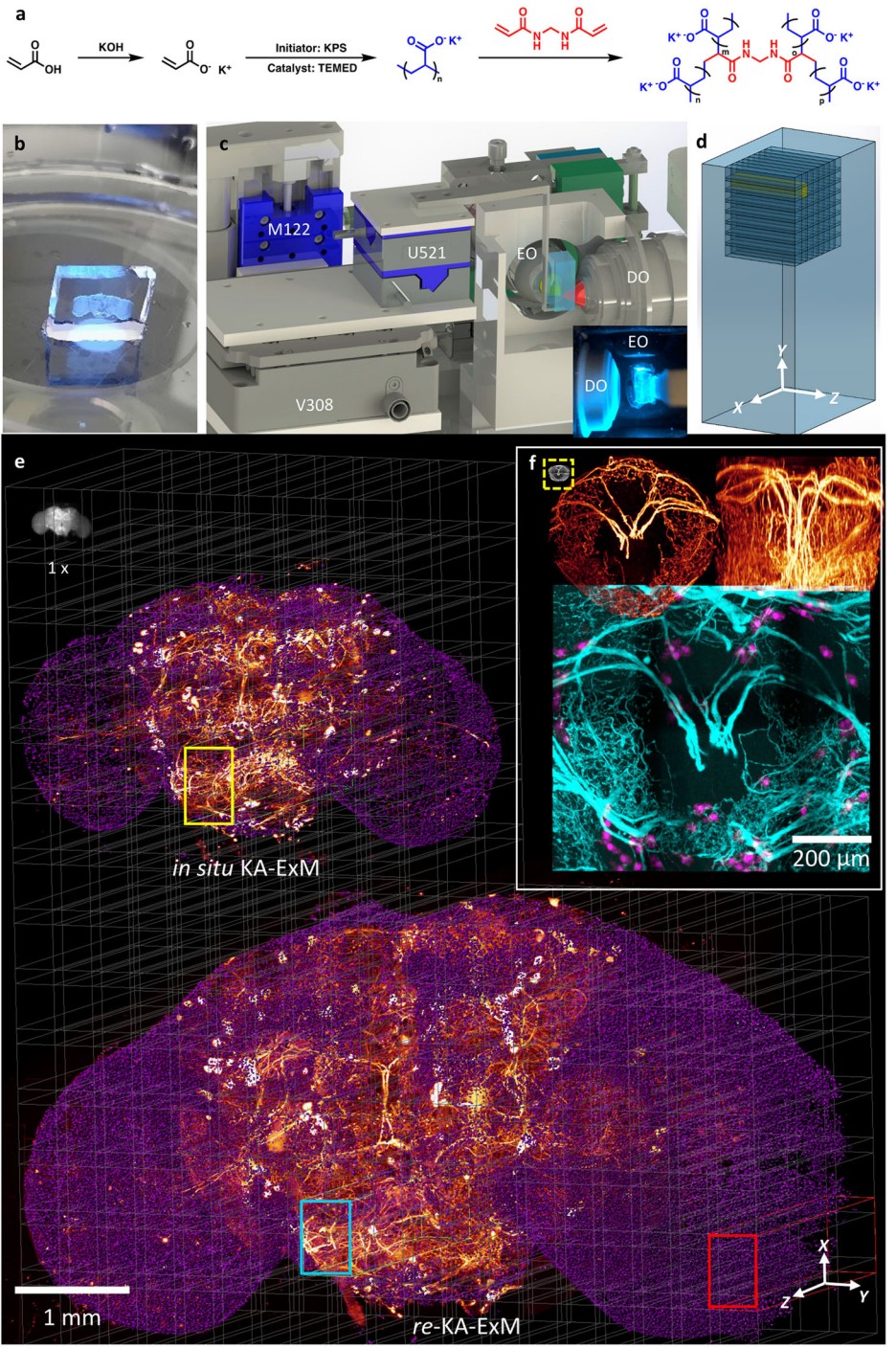

**Fig. 1 | KA-ExM with Bessel lightsheet microscopy (ΔBLX) for expansion and imaging of whole *Drosophila* brains. a** Synthesis of the in situ KA-ExM hydrogel. KPS potassium persulfate, TEMED tetramethylethylenediamine. **b** A free-standing KA hydrogel hosting an expanded fly brain stained with DAPI and illuminated by a 405 nm laser under a wide-field microscope. **c** The core ΔBLX system for imaging free-standing hydrogels comprises two ultrasonic piezo motors (U521) and one DC motor (M122) for sample translation, as well as a voice coil stage (V308) for sample scanning acquisition up to 7 mm. EO excitation objective, DO detection objective. Inset is a photograph of a centimeter-sized hydrogel in the water-filled imaging core. **d** Schematic for how a centimeter-sized gel is tiled by sample scanning using the voice coil motor. Note that there is no tiling in the z direction. **e** Maximum

intensity projection (MIP) images of a series of expansions acquired by ΔBLX from an identical fly brain (grayscale image) with GFP-expressing dopaminergic neurons (heatmap) and nuclear staining (purple) first subjected to in situ KA-ExM and then iterative expansion (*re*-KA-ExM). Five independent gel expansions performed with similar results. **f** Enlarged 3D images for the ellipsoid body (EB) neuropil extracted from the expanded whole fly brain images in (**e**): heatmap images are the top (left) and side (right) views from an in situ KA-ExM brain and the two-color image (cyan and purple) is a top view of the *re*-KA-ExM sample. Note that the size of the original EB is shown in the yellow dashed box and all images have the same scale bar. Gamma correction applied to display colors on the 3D rendering.

Notably, the actual iterative expansion ratio is 1.5 in the *x*, *y* and *z* planes even though we applied a 4x protocol[3]. Although neuronal and cellular features of the *re*-KA-ExM sample are maintained, their relative positions appear to be slightly altered due to localized non-uniform expansion (Supplementary Fig. 3). Note that this *re*-KA-ExM and GFP_635 immunostained *TH-GAL4* fly brain underwent almost 500,000 ΔBLX lightsheet exposures, indicating that our protocol is capable of maintaining the main biological features of the specimen whilst limiting photo-bleaching. Even though the huge expansion process resulted in considerable fluorophore dilution due to the rapid volumetric increase (more than 2,000-fold based on the 13x expansion ratio), detectable fluorescence signals were retained for both in situ KA-ExM and iterative *re*-KA-ExM.

To visualize the detailed structures, we present different views of a selected neuropil, i.e., an ellipsoid body (EB), in the central complex of the fly brain (Fig. 1f). The small grayscale image in the upper left panel is the original EB (~50 μm), and top- (upper left) and side-view (upper right) images of the EB with labeled dopaminergic neurons (~400 μm, heatmap) following in situ KA-ExM are also shown. In the bottom panel of Fig. 1f, a two-color volumetric image (showing dopaminergic neurons in cyan and nuclei in purple) of the millimeter-sized EB following the *re*-KA-ExM protocol is presented.

### Validation of biostructural integrity by pre- and post-iterative KA-ExM 3D imaging

To further explore fine structures, selected optical slices of the fly brain illustrated in Fig. 1e are shown in Fig. 2a–c. We focused on the prow (PRW) neuropil located in the subesophageal zone, which responds to sugar and bitter tastants (highlighted in Fig. 2a–c by dashed boxes to demonstrate the spatial resolution achieved by the expansion processes)[25,26]. For comparison, we imaged the optically cleared unexpanded *TH-GAL4* fly brain using a confocal system (Fig. 2d) to resolve the punctate expression pattern (laterally and axially) of this unexpanded dopaminergic neuron (zoomed in red dashed box in Fig. 2d), where the detailed bulbous synapse-like structures could not be resolved due to the diffraction limit. Figure 2e, f are MIP images recorded for the in situ-KA-ExM and *re*-KA-ExM shown in Fig. 1e (yellow and blue boxes), together with the corresponding dashed boxes presented in Fig. 2b, c to compare identical regions of both expanded samples. Note that the same scale bar is presented in Fig. 2d–f for the same neuropil, with further enlargements of the punctate patterns (white dashed boxes) shown for the same dimensions in Fig. 2g, h. The areas marked in Fig. 2i, j demonstrate that the same expanded synapse-like structures (20 μm or ~1 μm effective size) may represent specialized nerve endings, allowing 2D visualization of nanostructures with an effective pixel size of ~25 nm after 13x expansion (such as the space inside a presynapse, white arrow in Fig. 2j). Note that the white scale bars for all panels are in real physical units while the yellow scale bars indicate the actual size of the image divided by the expansion factor. Therefore, we show that nanoscale fly brain biostructures are retained despite undergoing iterative expansion and being imaged using a NA = 0.6 objective lens. More quantitative evaluations of the increased resolution we achieved based on full-width at half maximum measurement for pre- and post-expanded fly brain biostructures are presented in Supplementary Fig. 4.

### ΔBLX 3D imaging of PKA-ExM and *re*-PKA-ExM outcomes to visualize sub-synaptic structures

To increase the expansion ratio, we used commercially available potassium polyacrylate (PKA) oligomers to react with DMAA as gel building blocks (Fig. 3a). Note that DMAA self-reactions also occurred. Due to the DMAA, the resulting PKA gel was slightly more watery than the KA gel, enabling an even greater expansion ratio of ~15x, as determined from epi-fluorescence imaging of seven fly brains pre- and post-expansion (Supplementary Fig. 5 and Supplementary Table 4). To better resolve this PKA-ExM brain in 3D, we subjected it to ΔBLX (Fig. 3b and Supplementary Table 3, original fly brain size is shown in grayscale). In Supplementary Movie 3, we present a movie of the entire PKA-expanded brain, showing sliced views imaged by ΔBLX at *Z* intervals of 3 μm. In this case, a smaller expansion factor of ~10x was obtained, likely attributable to long-term exposure to the 3 M mounting adhesive. Again, we selected the PRW neuropil for further examination (Fig. 3b), with deconvolution at a *Z* step of 1 μm covering a thickness of ~2 mm (Fig. 3c). The sub-volumes highlighted by green and blue dashed boxes (Fig. 3c) have been enlarged and rendered as MIP images in the corresponding colored frames at right, with the green frame displaying punctate pre-synaptic structures (indicated by a white arrow) of similar size to those presented in Fig. 2j. The fine dopaminergic neuronal fibers (<100 nm in width, indicated by a yellow arrow, Fig. 3c) could be imaged upon ~10x expansion, allowing us to visualize empty spaces in the neural fibers not occupied by TH cytosolic signals (perhaps organelles, indicated by a blue arrow, an enlarged slice view in the inset of Fig. 3c).

Following iterative PKA gel expansion, we obtained even better spatial resolution to support that supposition (Supplementary Fig. 6a, b), with the area near the fly brain central complex presenting very detailed neural structures and good fluorescent signals. In Fig. 3d, we present iterative PKA (*re*-PKA ExM) images with a *Z* interval of 2 μm (covering a thickness of 3 mm) (see also Supplementary Fig. 6a, b). Within the blue frame of Fig. 3d, cell bodies have been enlarged to ~250 μm. The sub-volume framed in yellow in Fig. 3d shows the enhanced resolution of pre-synaptic boutons (white arrow), enabling better visualization of cellular organelles (empty spaces likely representing mitochondria) inside the pre-synaptic boutons compared to the images shown in Fig. 2j. Based on FlyWire and electron microscopy data on PRW.55 dopaminergic neurons (Supplementary Fig. 6c), the sizes and shapes of those "empty" spaces are consistent with their being occupied by mitochondria.

### Iterative monitoring of a 30x-plus *re*-PKA-ExM 3D volume of an identical fly brain

To compare the unexpanded and expanded samples, we present a series of images of the same fly brain sample subjected to nucleus staining and that then underwent PKA-ExM and *re*-PKA-ExM protocols (Fig. 4a and Supplementary Fig. 7a). The images are presented on a checkered canvas to facilitate size comparison. The original fly brain after PKA and DMAA treatment (1x), i.e., before any expansion, is ~540 μm in length (top left grayscale image). After treatment with digestion buffer (DB), the sample expanded to ~2.2 mm, i.e., already a 4x expansion (top middle grayscale image). Following a washing step with water, ~12x expansion was achieved (top right heatmap image), as determined by ΔBLX 3D imaging with a sample size of 6.8 x 3.4 x 2.5 mm³ composed of 114 tiles (Supplementary Table 3). After imaging by lightsheet microscopy, we removed this 12x fly brain sample from the imaging chamber and subjected it to a second gelation procedure (*re*-PKA-ExM) (Supplementary Fig. 7a). This *re*-PKA-ExM sample was so large it exceeded the current ΔBLX detection range. Therefore, we cut this ~2 cm-sized fly brain in half (shown in cyan and pink colors in Supplementary Fig. 8) so that the halved sample met the working criteria of ΔBLX. A 3D MIP image of this halved sample is shown in the bottom heatmap image of Fig. 4a, revealing an expansion ratio of 32x. This enormous halved fly brain image is comprised of 414 tiles and 500,000 lightsheet exposures, reflecting a physical size of 9.2 x 7.4 x 5.7 mm³, which was cut from a complete 32x-expanded fly brain with a size of 19.1 x 9.6 x 5.7 mm³ (Supplementary Table 3). Figure 4b shows the sliced view of the area marked in yellow frame in the 12x expanded fly brain in Fig. 4a, where cell density declined substantially within the same field-of-view (~500 μm) due to expansion of the sample volume.

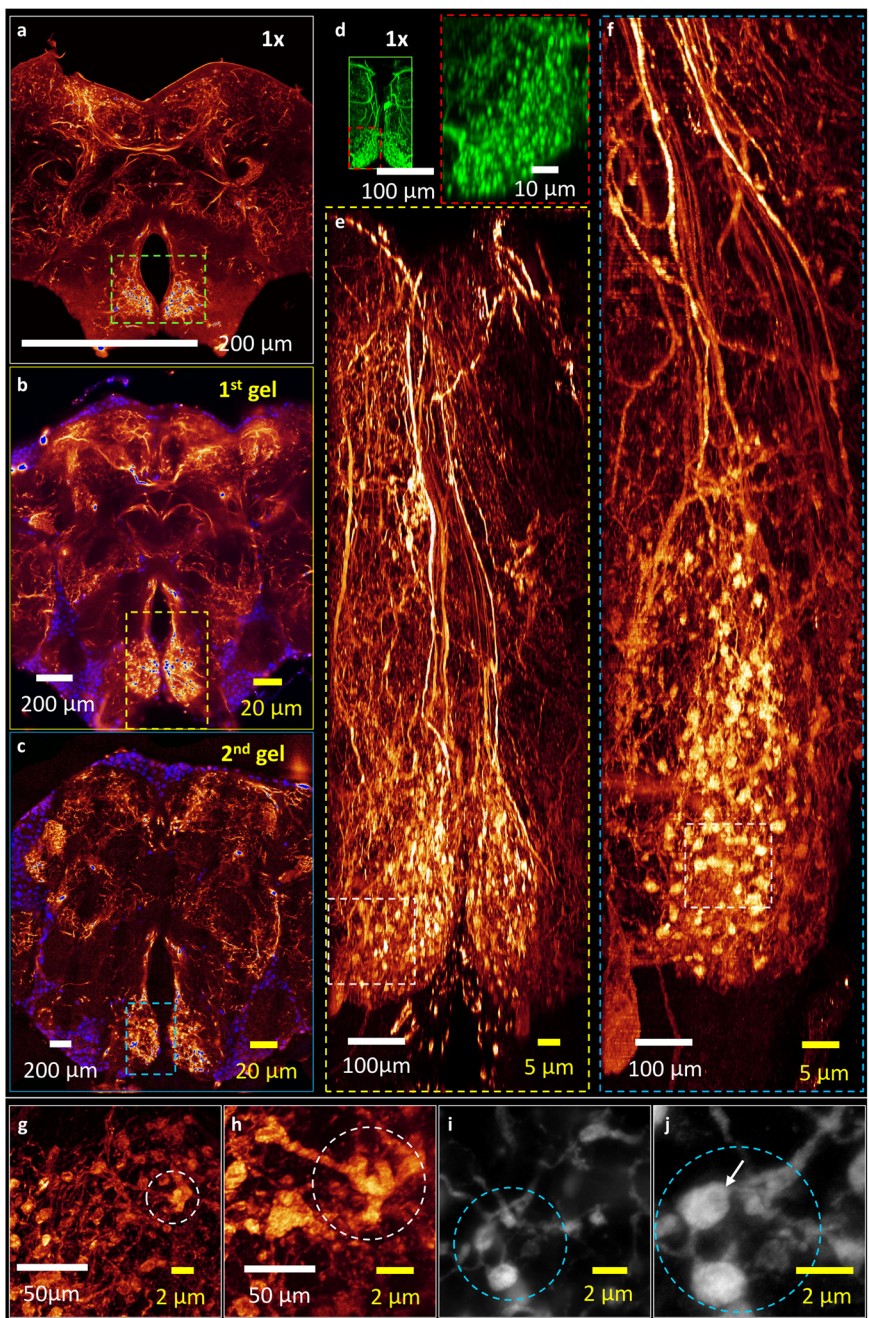

**Fig. 2 | Comparison of MIP images of the central nervous system of the *Drosophila* brain pre-expansion, and following in situ KA-ExM and *re*-KA-ExM.** **a** Confocal image of dopaminergic neurons where PRW neuropil marked in a green box in the original (1x) *TH-GAL4*, 20XUAS-6XGFP/+ *Drosophila* using a NA = 1.1 objective. **b**, **c** Two-color images (purple, nuclei; heatmap, dopaminergic neurons) of the identical *TH-GAL4* fly brain stained with DAPI shown in (**a**) after in situ KA-ExM (8x) and *re*-KA ExM (13x). The identical PRW neuropil highlighted in the yellow and blue dashed boxes for iterative expansion process. **d** A green box for confocal image of the partial area within the green dashed box in (**a**) and enlarged view of the punctate structures outlined by the red dashed box. **e**, **f** Enlarged views of the regions highlighted by the colored dashed boxes in (**b**) and (**c**), respectively, with the same scale bar for the same neuropil following in situ KA-ExM (8x) and *re*-KA ExM (13x). **g**, **h** Zoomed-in views of the selected areas within the white dashed boxes of (**e**) and (**f**). The marked areas (white dotted circles) show the same biological structures after 8x and 13x expansion. **i**, **j** Raw images of the same punctuate structures (highlighted in blue dotted circles) after in situ KA-ExM and *re*-KA-ExM. Note that the scale bars for all panels in real physical units (white font) vary according to the expansion factor (yellow font). Five independent gel expansions performed with similar results. Gamma correction applied to display colors on the 3D image rendering.

For comparing the same areas in the 12x- and 32x-expanded fly brain, slice views in Fig. 4c reveal the identical sizes of cell nuclei (red and green outlines) near the optic lobe within the whole brain (marked in white box in Fig. 4b) following iterative PKA-ExM protocols (inset: -12x, main image: 32x), with the nuclei being enlarged 3-fold by iterative expansion (-20 μm in length) and clearly revealing the internal DAPI

staining pattern. Moreover, for 3D volume rendering, Fig. 4d shows the nuclei have been segmented and rendered in different colors within the 3D image sharing the same scale bar, where the good separation between nuclei in the axial direction is a useful feature of such high sample expansion to resolve the inherent limitation in optical lens's axial resolving power.

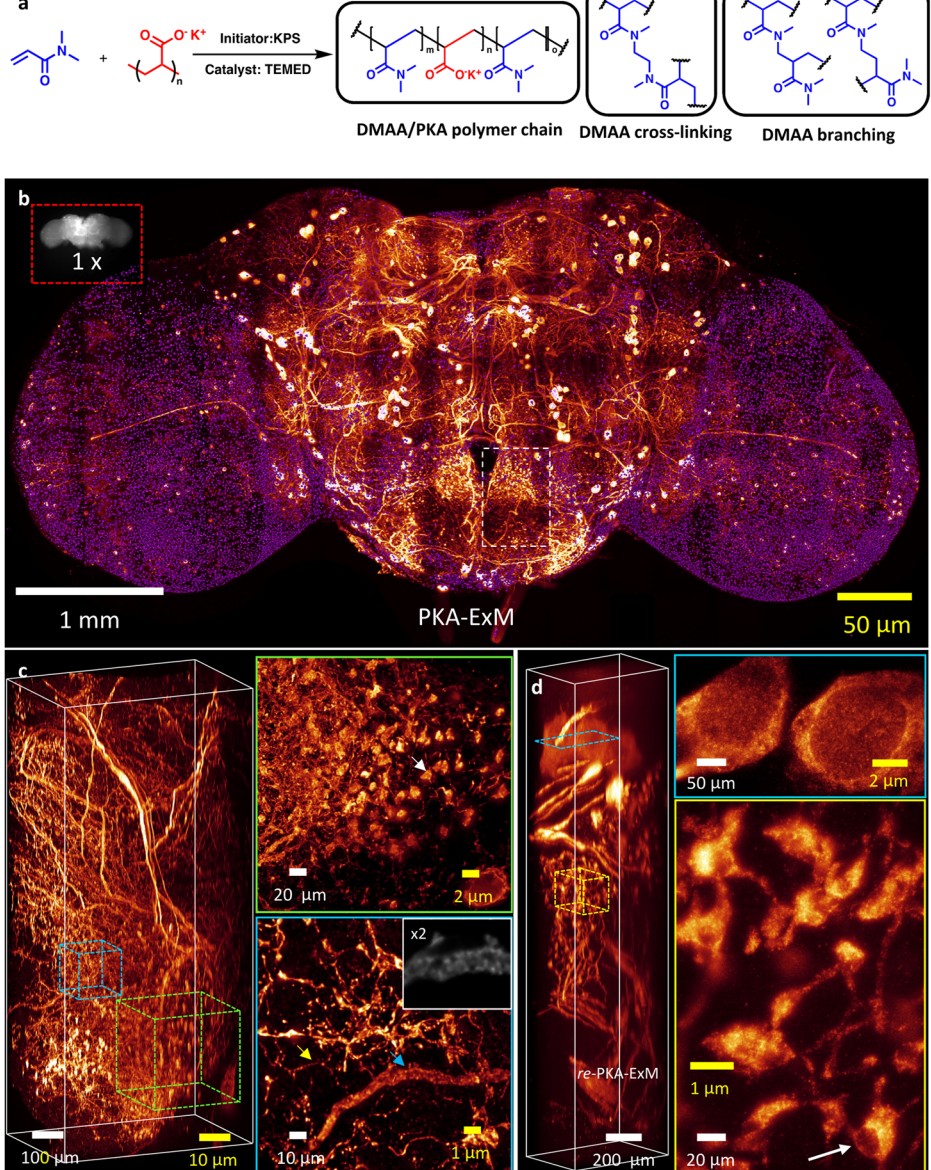

**Fig. 3 | Oligomeric PKA-ExM and respective iterative ExM (*re*-PKA-ExM) of an identical *TH-GAL4 Drosophila* brain. a** The PKA-based ExM chemical reaction of DMAA with potassium polyacrylate (oligomeric form) with cross-linking and self-crosslinking products, as initiated by potassium persulfate (KPS) and accelerated by tetramethylethylenediamine (TEMED). **b** The original (1x) *TH-GAL4*, 20XUAS-6XGFP/+ *Drosophila* brain (grayscale image, top left in a red box) and following PKA-ExM treatment giving rise to 10x expansion, as well as a two-color (purple, nuclei; heatmap, dopaminergic neurons) image of the whole 10x-expanded fly brain subjected to tile scanning by ΔBLX. **c** The tile demarcated by a white dashed box in (**b**) rendered in 3D side view to demonstrate axial resolution. The two sub-volumes highlighted by green and blue dashed cubes are shown at right, revealing punctate (white arrow) and fibrous neuronal structures (yellow and blue arrows). A 2x enlarged slice view marked with blue arrow shown in the inset. **d** Side-view in 3D of the same tile from (**b**) following *re*-PKA-ExM treatment, resulting in a 30x expansion ratio. A sliced view of dopaminergic neuronal cell bodies (~250 μm, blue frame) and a sub-volume illustrating puncta marked with white arrow (30 – 40 μm, yellow frame). Note that the scale bars for all panels in real physical units (white font) vary according to the expansion factor (yellow font). Seven independent PKA-ExM performed with similar results. Gamma correction applied to display colors on the 3D image rendering.

## 3D mapping of Tm5a visual neuronal processes in the *re*-PKA-ExM fly brain

Given these promising results for *re*-KA ExM and *re*-PKA ExM, we decided to examine sparsely labeled visual neuron Tm5a, which is challenging to visualize due to its complex dendritic processes spanning multiple *Drosophila* medulla layers. The *re*-PKA-treated fly brain (~2 cm), which had been labeled with membrane marker CD4-tandem dimer (td) GFP[27], is shown as an inset in Fig. 5a (outlined in red, with a red arrow pointing to the optic lobe). The main image of Fig. 5a represents a ΔBLX image of this expanded lobe (~7 mm), with *Z*-stack depth of neurons pseudocolored for 4.5 mm along the posterior-axial view, revealing the cell bodies (soma) and dendritic processes across the medulla. The enlarged Tm5a neurons are presented in Fig. 5b (heatmap image), for which the dendrite fields (red, green and yellow) and axon terminal (blue) of several segmented millimeter-sized Tm5a neurons are shown. A side view of these Tm5a neurons is illustrated in Fig. 5c, with each primary Tm5a dendrite serving as an epicenter from which lateral branches extend separately to neighboring columns in the M3, M6, and M8 medullary layers, (please

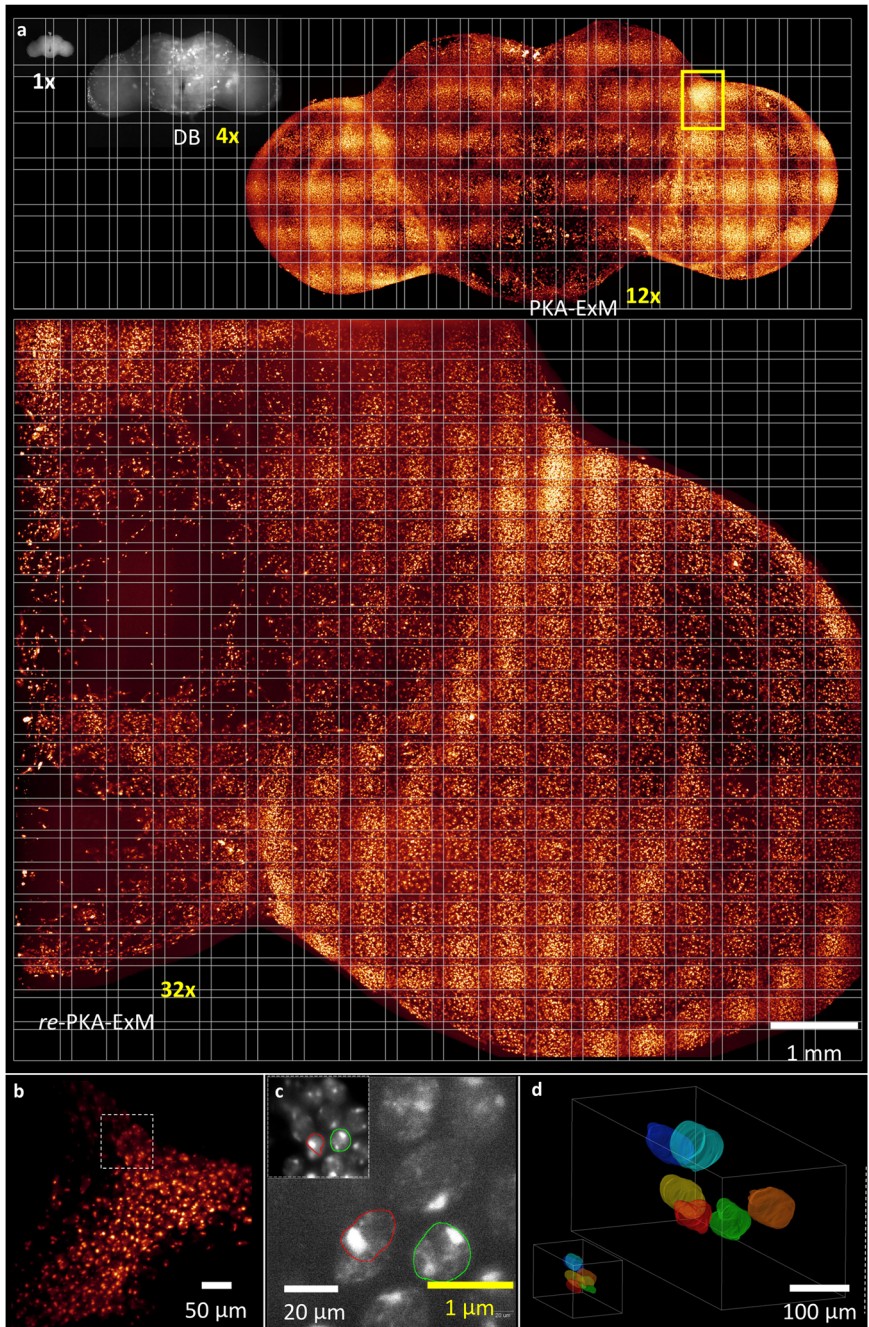

**Fig. 4 | ΔBLX imaging of an identical DAPI-stained *Drosophila* brain sample pre- and post- PKA-ExM expansion. a** All of the different expanded samples, i.e., 1x (grayscale, upper left), 4x (grayscale, upper middle), 12x (heatmap, upper right), and 32x (heatmap, bottom) of an identical fly brain presented on the same checkered canvas and subjected to 3D ΔBLX imaging. Note that the scale bar for all images is 1 mm. The area demarcated by yellow frame is used for comparing the 12x and 32x images. Three independent measurements performed with similar results. **b** A slice view for the area marked with yellow frame in (**a**) for 12x expanded fly brain. **c** Enlarged views in the white dashed box in (**b**) of the same nuclei colored red and green for PKA-ExM (12x, inset) and *re*-PKA-ExM (32x, main image), with the same scale bar for the inset and main image. **d** 3D renderings of the same selected regions in the dashed frame in (**b**), representing PKA-ExM (left) and *re*-PKA-ExM (right). In both cases, six nuclei have been segmented and rendered in different colors within the 3D image. Note that the scale bars in real physical units (white font) vary according to the expansion factor (yellow font). Gamma correction applied to display colors on the 3D image rendering.

refer to the confocal imaging presented in Supplementary Fig. 9a, b). We achieved an expansion ratio of >40x based on comparison of axon fiber diameter (expanded axon diameter of ~17 μm in Fig. 5c; unexpanded axon diameter of ~0.3 μm in the confocal image of Supplementary Fig. 9c). In Fig. 5d, a single Tm5a neuron (green) of several millimeters in length is presented, the dendritic field of which in the M6 layer spans ~250 μm. The yellow one in

Fig. 5b could be further segmented into two neurons. Also, in Fig. 5d are shown two Tm5a neurons (yellow and pink) located in very close proximity whose individual dendrites in the M6 layer are entangled (Supplementary Movie 4), an aspect that could not be discerned from confocal imaging data determined using a high-resolution NA = 1.1 objective lens (Supplementary Fig. 9d, e). However, our iterative expansion protocol and ΔBLX imaging

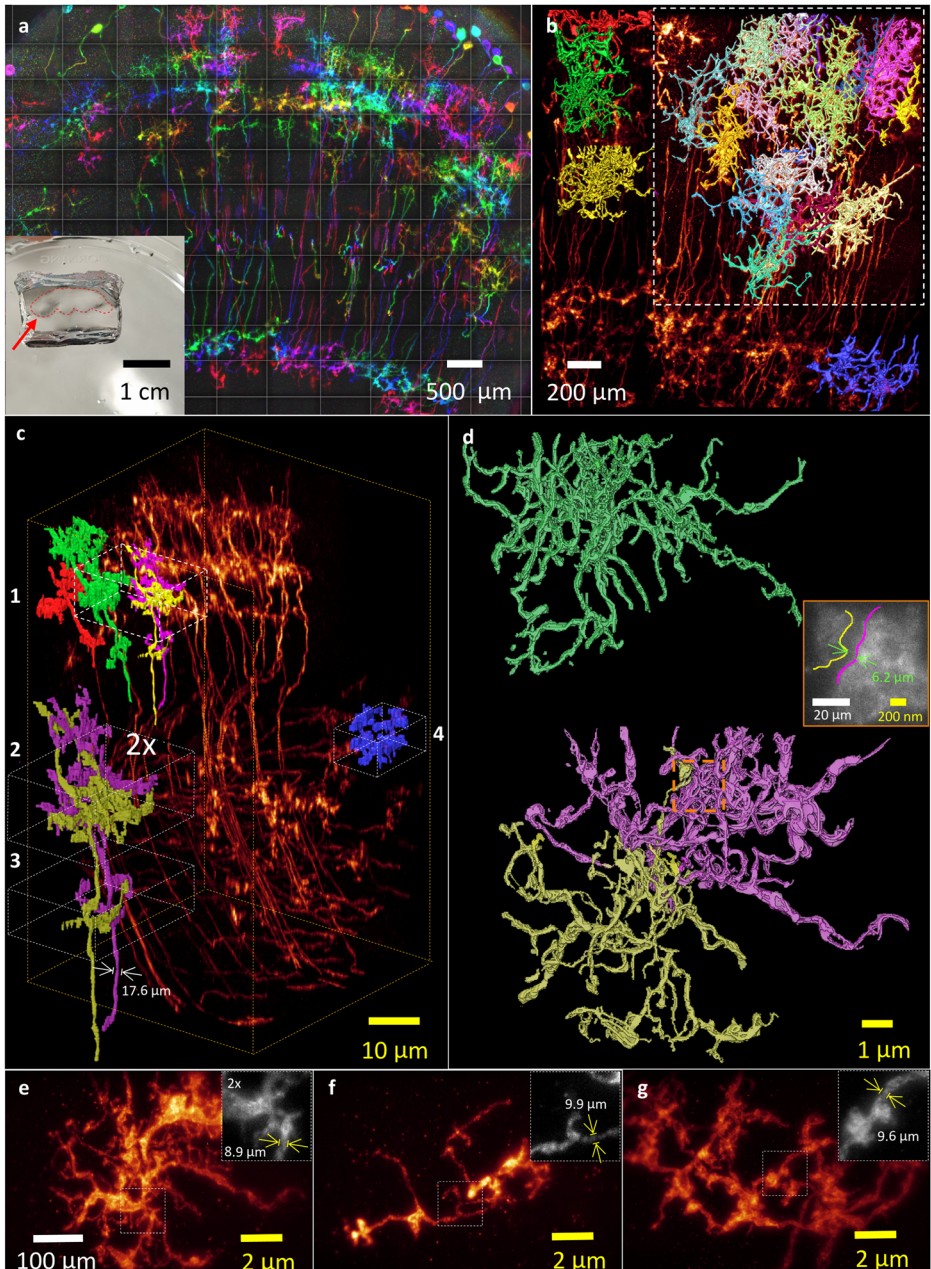

**Fig. 5 | Segmentation of Tm5a visual neurons in the *Drosophila* optic lobe following *re*-PKA-ExM expansion. a** MIP image colored according to depth of field showing the distributions of Tm5a neurons spanning multiple medullary layers. A hydrogel hosting an expanded fly brain marked with red arrow (outlined in red) is shown in the inset. Ten independent trials performed with similar results. **b** Enlarged MIP (heatmap) of a group of Tm5a neurons, with selected individual neurons highlighted in red, green and yellow, as well as a single axon terminal in blue. A group of segmented Tm5a neurons shown in the white dashed box. **c** 3D side view rendering of the data shown in (**b**), again showing the four selected Tm5a neurons and one axon terminal. An enlarged view (2-fold magnification) of two entangled Tm5a neurons (yellow and pink) is shown at bottom left. **d** Top view of the white dashed box labeled "1" in (**c**), representing the extent of the Tm5a neuron in the M6 layer. A zoomed-in slice view of two adjacent dendritic processes (outlined in an orange dashed box) is shown in the inset. **e**–**g** Top views of the white dashed boxes labeled "2", "3", and "4" in (**c**), representing the extent of the Tm5a neuron in the M6, M8, and terminal layers, respectively. The insets in (**e**), (**f**) represent 2-fold magnifications of the regions demarcated by white dashed boxes in the respective main images. Note that the scale bars for all panels in real physical units (white font) reflect the 40x expansion factor (yellow font). Gamma correction applied to display colors on the 3D image rendering.

enabled us to distinguish these two intricately intertwined Tm5a dendrite processes, even with gaps of only ~7 μm (~20 camera pixels, as shown in the inset of Fig. 5d and Supplementary Fig. 10), which is equivalent to ~175 nm given an expansion factor of ~40x. Notably, as revealed by the blue-colored outlines in Fig. 5b, c, Tm5a axons terminate in the lobula layer. Details of Tm5a neural processes in the M6 and M8 layers are highlighted in

Fig. 5e, f, where the zoomed-in views (dashed boxes) shown in the insets illustrate the fine dendritic fibers of width ~10 μm, which is equivalent to ~250 nm given the 40x expansion factor. These optical resolutions are comparable to those obtained by electron microscopy (EM) and presented in neuPrint and FlyWire resources[28–31] (see examples in Supplementary Fig. 11). Indeed, the optical images presented in Fig. 5d reveal comparable profiles as

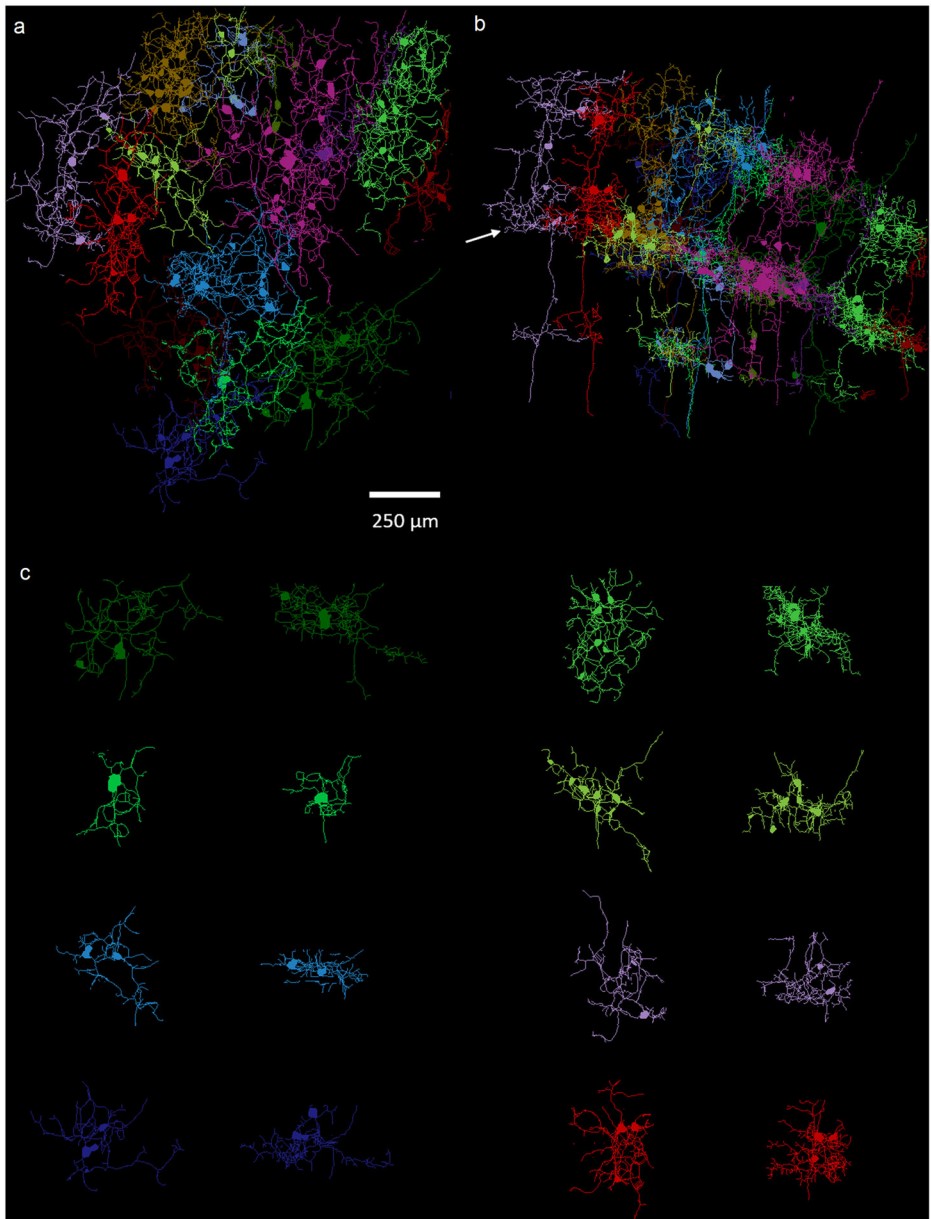

**Fig. 6 | 3D skeletonization feature of densely packed Tm5a neurons in the *re*-PKA ExM fly optic lobe with 40 plus expansion ratio. a** Top view of a group of skeletonized visual neurons Tm5a based on the segmentation data in the white dashed box of Fig. 5b. **b** Side view of these transmedullary neurons in (**a**) with their dendrites extending into the M3, M6, and M8 layers from top to bottom. **c** The top view (left) and side view (right) of the individual and colored Tm5a neurons with dendritic processes in M6 layer marked with white arrowhead in (**b**). Note that all the images sharing the same scale bar.

in the EM-imaged Tm5a neuron shown in Supplementary Fig. 11a. In the case of the Tm5a axonal terminal (Fig. 5g), the short processes and knotted structures of similar diameter to neuronal fibers are similar to profiles visualized by EM (Supplementary Fig. 11b), with the knotted structures likely being occupied by mitochondria. Moreover, a group of densely packed Tm5a neurons could be segmented in a largely expanded fly brain as displayed in the white dashed box in Fig. 5b, where ~15 neurons are colored and skeletonized as shown in Fig. 6. The top and side view of the skeletonized expanded Tm5a neurons with colors in Fig. 6a, b, respectively. The adjacent Tm5a neurons spanning lateral branches extend to reach neighboring columns in M3, M6, and M8 layers. Figure 6c shows top and side views of the representing segments for the dendrite field in M6 layer (white arrow in Fig. 6b). Supplementary Fig. 12 and Supplementary Movie 5

demonstrating the complete segmentation and various dendritic processes in M6 layer.

## 3D multicolor nano-resolution rendering of the L3 neuron and photoreceptors by ΔBLX

Given the merits of multi-color fluorescence detection in light microscopy, visualization of multiple targeted fluorescent proteins within the same expanded sample is possible for unraveling fine structures such as triple labeling to visualize cell nuclei (DAPI), photoreceptors R7 and R8 (GMR-RFP), and sparsely labeled visual neurons in a *Drosophila* optic lobe, as shown in Fig. 7a and sideview in Supplementary Fig. 13a. The volumetric imaging of an isolated photoreceptor in Supplementary Fig. 13b, c illustrate detailed R7 and R8 axon structures. In Fig. 7b and Supplementary Fig. 13d, e, we show a representative multi-color volumetric image of one segmented lamina monopolar neuron L3

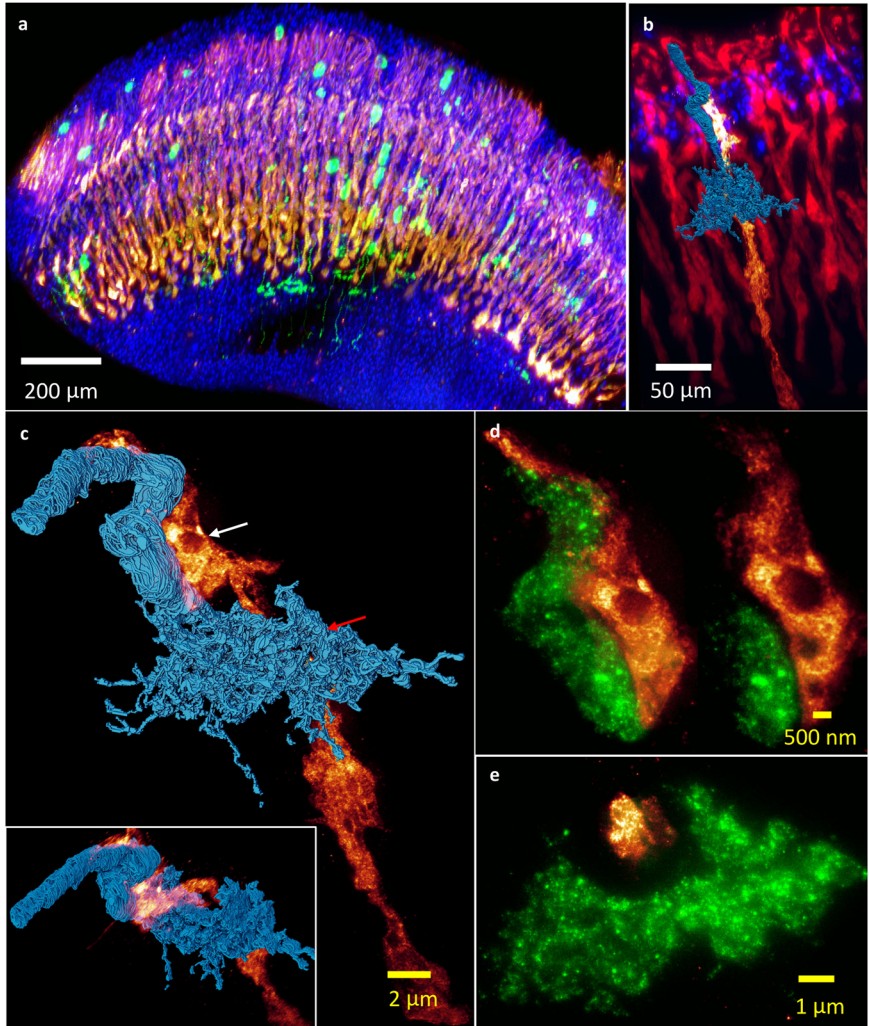

**Fig. 7 | 3D rendering of multicolor volumetric imaging of an expanded _Drosophila_ optic lobe with sparse labeling on lamina or medullary visual neurons.**
**a** Three-color MIP image for cell nuclei (DAPI) in blue, photoreceptors R7 and R8 (GMR-RFP) in red, and visual neurons (GFP-labeled membrane) in green. **b** A 3D view of the nuclei, photoreceptors, and one segmented lamina monopolar neuron, L3. **c** An enlarged view of the spatial arrangement of a L3 neuron (blue color) and photoreceptor (heatmap) pair, with a rotated view (clockwise) shown in the inset.

**d** Two slice images of the areas indicated by the white arrow in (**c**), with the photoreceptor shown in heatmap and L3 neuron axonal shafts in green. **e** A slice image of the L3 axon terminal, together with the photoreceptor (R7 and R8) area indicated by the red arrow in (**c**). Note that the scale bars in real physical units (white font) reflect the 40x expansion factor (yellow font). Gamma correction applied to display colors on the 3D image rendering.

(membrane labeled with GFP, shown in blue) among photoreceptors (in red). L3 has been identified as being responsible for luminance detection[32], receiving histaminergic outputs mainly from photoreceptors (R1 ~ R6 in lamina) and providing input to Tm9, Mi1, Mi9, and Tm20[33,34]. During development, both axon terminals of R8 and L3 target the M3 layer[35], as shown in Fig. 7c and by EM data in Supplementary Fig. 13f. The sliced volumetric image of the photoreceptor and L3 shown in Fig. 7d, e clearly illustrate the close proximity between axonal shafts of photoreceptors and L3 around the M1 ~ M2 layers (indicated by a white arrow), but exhibit an obvious separation in the M3 layer (indicated by a red arrow). According to EM data from NeuPrint (Supplementary Fig. 13g–i), the Dm9 neuron presumably fills up this space during development.

## Visualizing mitochondria in synaptic compartments and the scaffold protein Bruchpilot in the brain

In order to validate the surmised empty spaces occupied by mitochondria in Figs. 2j and 3d based on the lightsheet expansion imaging and EM data, a UAS-mCherry.mito.OMM/+; _TH-GAL4_, 20XUAS-6XGFP/

+ fly was subjected to PKA-ExM and imaged by ΔBLX. A representative image of the central area of the fly brain is presented in Supplementary Fig. 14a, with an enlargement of the two-color image of the PRW area shown in Fig. 8a, in which mCherry (red) specifically labels mitochondria enclosed within the GFP-labeled (green) presynaptic boutons. An expanded view of the red box in Fig. 8a is presented in Fig. 8b, showing a slice view of the rod-like mitochondria (white arrow), demonstrating the good resolving power of PKA-ExM and ΔBLX for a large field of view. As further evidence for the high spatial resolution gained from potassium (poly)acrylate-based ExM and lightsheet microscopy, we imaged the molecular architecture of synaptic active zones based on the nanoscopic distribution of the scaffold protein Brp labeled by a monoclonal antibody (Brp[Nc82]) in the adult fly brain[11,36]. The central complex and accessory neuropils are shown in Fig. 8c, with a fan-shaped body (FB) of ~3.5 mm marked in yellow. Note that the original size of the FB is about 80 ~ 90 μm[37], revealing a ~50x plus expansion ratio through _re_-PKA-ExM. 3D volume rendering of the asymmetrical body (AB) is also shown in blue in Fig. 8c[38], constructed by 2000 x 2600 x 301 pixels at a pixel resolution of 325 nm x 325 nm x

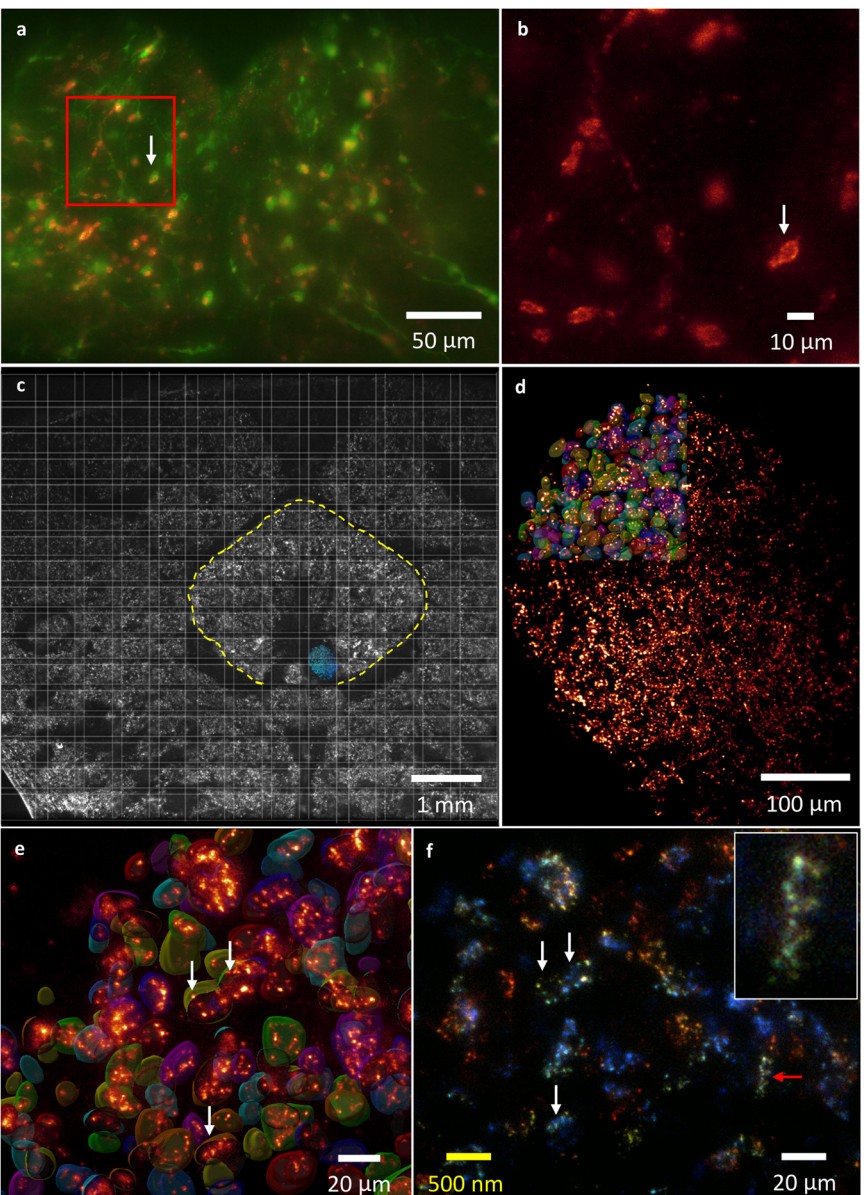

**Fig. 8 | 3D mapping of mitochondria inside presynaptic compartments and the nano-organization of Brp in the adult fly brain. a** Two-color MIP image of GFP-labeled *TH-GAL4* (green) and mCherry-labeled mitochondria (red) in the PRW region of the PKA-ExM fly brain. **b** Expanded slice view of the red box in (**a**) showing only mitochondria. The white arrow marked as identical presynaptic compartment in (**a**) and (**b**). Five independent measurements performed with similar results. **c** A tiled lightsheet image of the central complex of the *re*-PKA ExM fly brain, showing an expanded FB (yellow) of ~3.5 mm and the AB region highlighted in blue. Three trials made with similar results. **d** MIP image of the AB stained with Brp^Nc82 coupled with Cy3, with individual active zones highlighted in the top left part by surface rendering. **e** Enlarged 3D view of the identified active zones in the AB shown in (**d**), with different orientations of the T bar, the primary unit of active zone architecture (arrowheads). **f** Color-coded depth mapping of (**d**) to 30 μm with *z*-intervals of 10 μm. Single or multiple clusters in different orientations (arrowheads) are shown. The inset is a zoomed in view of the clusters marked by a red arrowhead. Note that the scale bars in real physical units (white font) reflect the 40x expansion factor (yellow font).

5 μm selected from the whole-brain image. An enlarged MIP view of the C-terminal epitope (Brp^Nc82) in combination with Cy3 labeling is presented in Fig. 8d, and a series of slice views at a *z* interval of 5 μm is shown in Supplementary Fig. 14b. Note that the boundary for intensity variation is caused by image stitching. We segmented ~261 Brp clusters in the upper left part of the image (highlighted by surface rendering). Magnifications of the identified active zones (Brp clusters) are displayed in Fig. 8e, f. Note that the scaffold clusters (arrowheads), a major and essential constituent of T-bars, display different orientations and are either separate or grouped. Moreover, color-coding of the Brp clusters in Fig. 8f could be mapped optically to a depth of

30 μm due to the highly expanded nature of the Brp scaffold, as shown in the inset of Fig. 8f.

For sample diversity, the validation of *re*-KA and *re*-PKA ExM on the spheroids generated from the HCT 116 cell line was performed. In Supplementary Fig. 15a, b, we present the expansion ratio of ~100 μm spheroids before and after expansion, resulting in ~20x for *re*-KA and ~40x for *re*-PKA, respectively. Figure 9a shows the image of original spheroid and Fig. 9b displays the expanded spheroid has ~4 mm size with maintained morphology after *re*-PKA ExM with its original size shown in the inset at the same scale. An enlarged 3D rendering and slice view for a mitotic cell (>300 μm) inside the spheroid shown in

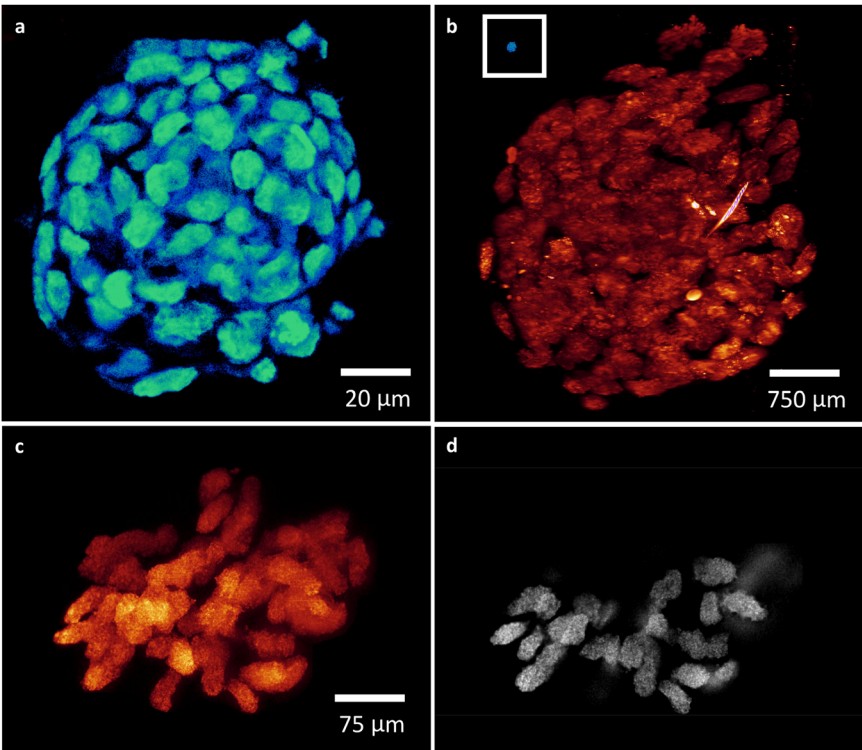

**Fig. 9 | The *re*-PKA ExM based on 100 µm spheroids generated from the HCT116 cell line in which nuclei stained by DAPI and imaged by ΔBLX. a** Image of the original ~100-µm spheroid. **b** Lightsheet imaging of the spheroid shown in (**a**) after the *re*-PKA-ExM process, resulting in a ~40x-plus expansion to ~4 mm (the original size is shown in the inset at the same scale). **c** Raw 3D MIP image of one ~300 µm mitotic cell selected from (**b**) after *re*-PKA-ExM. **d** Single slice view of (**c**) showing the detailed structure of the chromosomes. Five independent trials of *re*-KA and 4 trials of *re*-PKA ExM performed. Source data are provided as a source data file.

Fig. 9c, d with detailed chromosomal structures or an interphase one in Supplementary Fig. 15c, d.

## Discussion

Overall, it is quite challenging to maintain biostructures inside large watery gel-supported tissues upon increasing the expansion ratio of ExM. Here, we have shown that our KA- and PKA-based ExM protocols that are compatible with GA fixation retained such structures within fly brain samples, despite extensively expanding their physical sizes (see Fig. 2, Supplementary Fig. 16 and Supplementary Table 5 for summaries of in situ KA-ExM and PKA-ExM). Moreover, we measured a Young's modulus (ratio of tensile stress to tensile strain, reflecting the ability of a material to stretch and deform) of ~30 kPa for the in situ KA hydrogel (expansion factor of 8x), i.e., similar to that of muscles, and a Young's modulus of ~2000 kPa for a 4x ExM sample, i.e., similar to that of silicified diatoms[39,40]. However, we could not measure the Young's modulus for a 15x PKA-ExM hydrogel using the same liquid-phase atomic force microscope probe (see Supplementary Table 6). Another improvement achieved herein is that with strong dialdehyde fixatives such as GA or glyoxal, KA-based ExM lightsheet shows the capability of imaging preserved transgenic fluorescent proteins of interest within large expanded samples. Considering the dilution effect on the fluorophore intensity, stability of the chemical reagents and sample handling of the expanded tissues, in situ KA-ExM is preferable for most daily routine experiments. Given that spatial resolution is often critical, optimizing PKA-ExM could be the best option for increasing fluorescent signals, especially by altering experimental conditions such as the imaging buffer. Since PKA-ExM products, e.g., the superabsorbent polymers, become watery and very sensitive to the environmental ionic strength, diluted phosphate-buffered saline (~0.05%) is used during 3D lightsheet acquisition in order to maintain the sample

expansion ratio. In addition, the KA-based ExM could be applied extensively to large organs in other animal models such as spheroids[41], zebrafish or mouse[42]. Coupled with iterative expansion, KA or PKA hydrogels could be further expanded to achieve ~30x plus expansion depending on KA/PKA purity.

The neuronal processes of the *Drosophila* nervous system are remarkably delicate, typically measuring ~50 nm but occasionally being as small as 15 nm[43]. Consequently, the short working wavelength of EM endows advantages in visualizing even the finest of axonal branches. However, labor-intensive and time-consuming image mapping renders EM unsuitable for high-throughput analyses of large-scale tissues, especially when endeavoring to link molecular data to the intact brain[44–46]. Alternatively, in order to meticulously delineate the intricacies of dendritic processes or synaptic structures with whole brain coverage, a greater expansion ratio is required, resulting in rather bulky samples. To record the entire expanded brain at synaptic level without hindering subsequent imaging reconstruction, herein we present an imaging platform (ΔBLX) for visualizing at high resolution largely expanded samples (centimeter-sized) using KA/PKA hydrogels and Nyquist-based sampling of a long working distance NA = 0.6 imaging objective. Given an achievable expansion ratio of 40x, a pixel size of 325 nm corresponding to 8 nm is obtained for mapping fluorescence signal in KA/PKA-based ExM tissue samples. This technological leap not only facilitates precise tracing of the most minuscule dendritic processes, but also promises to unravel the complexities of neural connectivity.

Imaging at such a high isotropic expansion ratio upon formation of the very thick hydrogel prevents high NA objectives from directly accessing the sample, which is especially important when the aim is to view the entire expanded sample[47–49]. Hence, we had to use a lower NA = 0.6 with a long working distance to cover the whole expanded fly

brain. Nevertheless, using a NA = 0.6 lens still enabled us to view organelle compartments within synapses across the entire fly brain, which would not be possible without ExM. Since the resolving power of the NA = 0.6 lens is ~0.5 μm, depicting organelles within micrometer-sized synapses is typically curtailed by the limited numbers of optical imaging pixels in non-expanded samples. However, since the largely expanded 3D synapses we achieved (~20-30 μm) cover approximately 100 imaging pixels with an effective pixel size of ~12 nm in each dimension, we could observe more detailed structures within them. Last but not least, multicolor KA/PKA-based ExM enables comprehension of the complexities of cellular function and organization, such as the synapse formation between photoreceptors and a single lamina monopolar neuron L3 shown in Supplementary Fig. 17a. Moreover, it seems that R7 and R8 form limited synapses specifically away from the M3 layer with the L3 neuron. Instead, L3 associated with the Dm9 terminal, as shown in Supplementary Fig. 17b. These features could be further characterized and quantified by multicolor lightsheet microscopy with KA/PKA-based ExM.

Significantly, the sample contrast from specific fluorescence-tagged proteins of interest achievable through KA/PKA-based ExM could provide critical identification and characterization data for rich molecular phenotyping datasets arising from EM connectome studies. Moreover, by providing the means to explore dendritic morphogenesis across diverse genotypes, sparse labeling KA/PKA-based ExM can facilitate comparative analyses between wild-type and mutant counterparts, which are beyond the reach of volume-based EM connectome reconstructions. Using the segmented dataset for a highly expanded (>40x) fluorescence-labeled sample obtained using our system as a ground truth may enable application of deep learning processes to map dense neuronal segmentation over a large area (as exemplified in Fig. 6). By means of a greater sample expansion ratio from iterative process and axicon-based Bessel lightsheet to gain effective spatial resolution, imaging the nanoscopic molecular architecture of features such as the synaptic active zone, T-bar shown in the EM of Supplementary Fig. 17b, within the whole brain (as exemplified in Fig. 8 and Supplementary Fig. 14b) is now possible.

## Methods

### Axicon-based Bessel lightsheet microscope (ΔBLX)

A schematic of the optical system is presented in Supplementary Fig. 2. The scanning beam emanated from a laser combiner equipped with 405 nm (250 mW, RPMC, Oxxius LBX-405-300-CIR-PP), 488 nm (300 mW, MPB Communications, 2RU-VFL-P-300-488-B1R), 560 nm (500 mW, MPB Communications, 2RU-VFL-P-500-560-B1R), and 642 nm (500 mW, MPB Communications, 2RU-VFL-P-500-642-B1R) lasers. The lasers were combined with long-pass dichroic filters and aligned collinearly before entering an acousto-optical tunable filter (AOTF, AA Quanta Tech, Optoelectronic AOTF AOTFnC-400.650-TN), which we used to control laser exposure time, intensity and wavelength. Before passing through a half wave plate (Bolder Vision Optik, BVO AHWP3) and the AOTF, the laser beams were expanded to a 1/e2 diameter of 2.5 mm via two lenses (8 mm FL/12.24 mm Ø, Thorlabs C240TME-A; 20 mm FL/12.5 mm Ø Edmund 47-661) (L1 ~ L8). Following the AOTF, a pair of L9 and L10 lenses (60 mm FL/25.4 mm Ø, Thorlabs, AC254-060-A; 200 mm FL/25.4 mm dia, Thorlabs, ACY254-200-A) were used to expand the beam profile further as input for axicon-based Bessel beam generation. A combination of an L11 lens (250 mm FL/ 25.4 mm Ø, ThorLabs, ACY254-250-A) and an axicon lens (Thorlabs, AX2505-A) was used to form a focus laser ring, magnified 1.25X through relay lenses L12 and L13 (100 mm FL/25 mm Ø, Edmund NT47-641; 125 mm FL/25 mm Ø, Edmund NT49-361) in a 4 f arrangement, linked to a scanning system composed of two 5 mm galvos (GM, Cambridge Technology, 6215H) and one pair of matched focal length achromatic L14 and L15 relay lenses (75 mm FL/25 mm Ø, Edmund NT47-639; 75 mm FL/25 mm Ø, Edmund NT47-639) in a 4 f

arrangement. Since each galvo was conjugated to the back pupil of the customized excitation objective, this system enabled beam scanning along both the x and z axis of the sample. After passing through the scanning system, the image of the focus laser ring was re-magnified 2.67X through relay lenses L16 and L17 (150 mm FL/25 mm Ø, Edmund NT49-285; 400 mm FL/75 mm Ø, Edmund 88-595). The magnified laser ring was conjugated to the back focal plane of a custom-manufactured excitation objective (numerical aperture (N.A.) = 0.5, working distance (WD) = 12.8 mm; NARLabs, TIRI, Taiwan). The length of the lightsheet generated by the axicon lens extended to >660 μm (outer N.A. = 0.17, inner N.A. = 0.16). The excited fluorescence of the expanded sample was collected using an orthogonally-mounted detection objective (Olympus, XLPLN10XSVMP 10X, 0.6 NA, 8 mm WD), the focal plane of which was coincident with that of scanning Bessel beam illumination. Expanded samples were mounted on an L-shaped sample holder and translated and tiled using two ultrasonic piezo motors (Physik Instrumente, PILine linear stage, U521) with a traveling distance of 18 mm, one DC motor (Physik Instrumente, microtranslation stage, M122) with a traveling distance of 25 mm, and a voice coil stage (Physik Instrumente, Voice Coil PIFOC Focus, V308) with a traveling distance of 7 mm. The fluorescence signal was then imaged via an emission filter onto an sCMOS camera (Hamamatsu, Orca Flash 4.0 v2 sCMOS) using a 350 mm tube lens (350 mm FL/30 mm Ø, OptoSigma DLB-30-350PM) to provide an overall magnification of 20X.

### Image acquisition, processing and rendering

The free-standing expanded fluorescently-labeled samples were glued onto the L-shaped holder (see Fig. 1b), translated via a triple-axis (x, y, z) stage composed of the two ultrasonic piezo motors and a single DC motor (see Fig. 1c) at multiple positions to image an array of sub-volumes in sequence, and z-scanning was performed by the voice coil stage (see Fig. 1d). The camera exposure, x-galvo, and voice coil stage were synchronized for sample scanning to form a 3D image data, i.e., a stack of 2D tiff series utilizing the wide-field detection unit of the camera. The acquired 3D volumetric 16-bit raw images were stored in our in-house cluster, which also served as the platform for initial data preprocessing (such as data down-sampling). This cluster consisted of a storage capacity of 300 TB, three computing nodes, an Infiniband network for high-throughput (54 Gb/s) intercommunication, and a head node as the gateway and control center, with Debian GNU/Linux 9.13 acting as the operating system. The storage system was operated by a Lustre-2.12.6 parallel file system, containing one meta-data server for indexing of files and directories, and two object data servers for parallel data input/output (I/O). Each computing node was equipped with dual Intel Xeon Gold 6230 graphics processing units (GPUs), comprising a total of 40 central processing unit (CPU) cores, and 512 GB memory. The Slurm-16.05 job queuing system was installed in the cluster for job and resource management. Finally, a 10 G fiber wire connected the head node and our laboratory network, ensuring efficient data movement between the cluster and our laboratory.

Our data down-sampling code was implemented in python-3.8 under the Anaconda3 environment. For the dataset of re-PKA ExM images shown in Fig. 4a, 18 ×23 x 1 tiles had to be stitched, with each tile consisting of 2048 ×1408 x 1226 pixels (x, y, z) at a voxel size of 0.325 μm x 0.325 μm x 3 μm. The unit volume was ~6.9 GB, divided in half for data storage so that each "chunk" was ≤ 4.0 GB. Total data size was calculated as approximately 2.66 TB. Accordingly, to enhance processing performance, we enabled multi-thread processing by adopting the python module dask-2.20 in our code, so that file chunks were distributed to each thread and handled in parallel. Image data were typically down-sampled to (1/8, 1/8, 1) or (1/4, 1/4, 1) in the x, y, z directions, respectively. To balance the bottleneck of data I/O and computing speed for more abundant threads, we used eight threads for each down-sampling task, resulting in approximately 2.5-3.5 times faster processing compared to running the same task serially.

Image stitching from file chunks was performed with the Imaris Stitcher software, Imaris 9.5 or Fiji plugin Big Stitcher[50,51], running on our Windows workstations equipped with dual Intel Xeon Gold 6256 CPUs (total 24 CPU cores, 1 TB memory, and 90 TB storage for working space). All imaging rendering and figure preparations were conducted in Autodesk Inventor, Amira 3D 2024.1 software (Thermo Fisher Scientific), Huygens Software for deconvolution and ITK-SNAP open-source software[52].

## Sample information

We have complied with all relevant ethical regulations. Fly stocks were raised on cornmeal food, and maintained at a temperature of 25 °C and 70% relative humidity under a 12-h light/dark cycle. The following fly lines were used in the current study: *TH-GAL4* (8848, BDSC); 20XUAS-6XGFP@attP2 (52262, BDSC); UAS-mCherry.mito.OMM (66532, BDSC); 13xLexAop2-CD4-tdGFP (77136, BDSC); OrtC1-3-GAL4, GMR-myr-mRFP (7121, BDSC), and Tm5a split-LexA (from T.Y.Lin and C.H.Lee). The following fly genotypes were generated based on the aforementioned fly strains: *TH-GAL4 > 20XUAS-6XGFP*, UAS-mCherry.mito.OMM for labeling mitochondria in dopaminergic neurons; Tm5a split-LexA>13xLexAop2-CD4-tdGFP to sparsely label Tm5a neurons; and a L3 single cell MARCM clone generated by OrtC1-3-GAL4 in the visual system.

Human colorectal cells HCT 116 (BCRC 60349, Bioresource collection and Research Center, Taiwan) were maintained in culture medium containing McCoy's 5 A medium (Gibco 16600, Invitrogen Co., Carlsbad, CA), 10% (v/v) fetal bovine serum (Gibco 26140-079, Invitrogen), and 1% (v/v) antibiotic-antimitotic (Gibco 15240, Invitrogen) at 37 °C in a humidified incubator with 5% $CO_2$. The stocks were maintained in a T25 cell culture flask (Nunc 156367, Thermo Scientific Inc., Rochester, NY), and passaged by dissociation with TrypLE™ Express Enzyme (1X) (Gibco 12604, Invitrogen).

For the 3D spheroid formation, HCT 116 cells were seeded into the U-shaped wells of the ultra-low attachment 96-well microplate (Costar 7007, Corning Incorporated-Life Sciences, ME 04043, USA) at a density of 50 cells per well in a volume of 100 µL. The 96-well plates were placed in the humidified cell incubator at 37 °C, with 95% air and 5% $CO_2$ atmosphere. After 3 days of incubation, the HCT 116 spheroids were fixed and processed with tissue expansion procedure.

## Fixation and immunostaining

All buffer components and chemical name abbreviations are detailed in Supplementary Table 7. Freshly dissected *Drosophila* brains were fixed with 4% PFA for 30 min on ice, before undergoing 0.2% GA fixation on ice under vacuum for another 20 min. After de-vacuuming by flow-through with nitrogen gas, the brains were washed three times with 1X PBS for 5 min each time. The fixed brain samples were permeabilized and blocked with blocking buffer for 1 hr at 37 °C with gentle shaking. Antibodies were diluted in dilution buffer. Samples were first incubated in primary antibody (1:200-1:100, GFP Antibody, Thermo Fisher A-11122; 1:50, Brp Antibody, DSHB Cat# nc82) for 3 days at room temperature with gentle shaking. Then they were washed three times with wash buffer for 15 min at room temperature, before being incubated in secondary antibody (1:200-1:100, Goat anti-Rabbit IgG (H + L) Secondary Antibody, Biotin, Thermo Fisher 65-6140; Goat anti-Mouse conjugate Cy3, Jackson ImmunoResearch 115-165-166) for 1 day at room temperature. After washing three times for 15 min each time with wash buffer, the samples stained with biotin conjugated secondary antibody were incubated in Alexa Fluo-635 conjugated streptavidin (1:200-1:100, Streptavidin, Alexa Fluor™ 635 conjugate, Thermo Fisher S32364) for 1 day at room temperature. Finally, the samples were washed three times for 15 min each time with wash buffer and post-fixed with 4% PFA with 0.2% GA for 20 min, followed by three 1X PBS washes for 15 min each time.

## Potassium acrylate (KA) or potassium polyacrylate (PKA) expansion microscopy

The anchoring treatments were performed in 10 mM MA-NHS overnight at 4 °C, and washed twice with 1X PBS for 5 min prior to a gelation step the next day. KA monomer stock was prepared according to one-step in situ synthesis. To make a total of 60 mL KA monomer stock, 11.2 g KOH was dissolved in 36 mL ddH2O, before adding 14.2 mL of acrylic acid (AA-liquid) into the completely dissolved KOH. When the KOH and AA were well mixed after stirring for 30 min, 17 mg MBA was added into the mixture and the total volume was adjusted to 60 mL. The resulting monomer solution was aliquoted for convenient use (450 µL per tube) and frozen immediately at -20 °C to maintain the monomer's freshness. PKA monomer stock was prepared by first dissolving 1.31 g PKA in 10 mL ddH2O, and then adding the mixture into an entire 5-mL pack of DMAA liquid, before adjusting the final volume to 16 mL with ddH2O. The monomer solution was aliquoted (450 µL per tube) and frozen at -20 °C immediately. To prepare ready-to-use gelling solution, a 450-µL aliquot of KA or PKA monomer solution was vacuumed for 10 min, de-vacuumed with N2 gas for 10 min, and this cycle was repeated once more. After placing a sample onto a 10-mm coverglass in a 3-cm petri-dish cushioned with a flat piece of parafilm, the gelling solution was mixed by adding 50 µL of stock KPS solution (0.036 g/mL in ddH2O) and 2 µL of 100% TEMED into the de-gassed 450 µL monomer right before use. The ready-to-use gelling solution (30 µl) was then added to the sample and covered with another 10-mm coverglass to make a sandwich-like gelation chamber. The 3-cm petri-dish containing the sample was then transferred into a covered humidified chamber, and gelation was performed at 37 °C for 2 hours. After sample gelation, the coverglasses were detached and excess gel was removed using a scalpel. The gel-embedded sample was then transferred to a small container with digestion buffer. Digestion was performed overnight at room temperature with gentle shaking.

The next day, digested KA gels were transferred into 6-cm petri-dishes and washed three times with 1X PBS for 15 min each time. Gels were stained using DAPI in 1X PBS (1:3000, Thermo Fisher 62248) for 4 hours at room temperature, and transferred into new 6-cm dishes three times (30 min each time) for water expansion, followed by overnight shaking in water at room temperature. Digested PKA gels were transferred into 6-cm petri-dishes for expansion in water for 10 min (once), then with DAPI (1:500 diluted in ddH2O, SouthernBiotech 0100-20) for 4 hours (once), with a final water wash overnight at room temperature with gentle shaking. The samples were expected to reach maximum extent by the following morning and could then be mounted for imaging by lightsheet microscopy.

## *re*-KA and *re*-PKA expansion microscopy

The iterative *re*-KA and *re*-PKA expansions began with the KA-expanded or PKA-expanded brains, respectively. The re-embedding and secondary gel processes were the same for *re*-KA and *re*-PKA expansion, unless otherwise specified. First, excess gel was removed using a scalpel, making sure to leave as little surrounding gel as possible. Typically, the KA-expanded *Drosophila* brain reached 5 ×3 x 2 mm in size, so the gel was cut to 6 ×4 x 3 mm using DAPI signal under a dissection microscope as a guide to sample location in the gel and for incisions. The resulting gel-embedded sample was washed twice for 10 min at room temperature in re-embedding wash buffer prepared freshly for each experiment, and then once again in the same re-embedding wash buffer for 1 hour. Then, the re-embedding gelling solution was prepared. The subsequent gelation procedure was similar to that described above for KA-ExM gels, using two coverglasses (12-mm Ø, Warner Ins. 64-0702-CS-12R) to make a sandwich chamber for gelation. Re-embedding gelation occurred at 45 °C for 2 hours.

After re-embedding gel polymerization, excess gel was again removed using a scalpel, and the gels were washed three times with 1X PBS for 15 min each time. For gel polymerization in advance of the

second expansion, samples were first washed twice (10 min each time) with the second gelling solution, and then once with the same gelling solution for 1 hour at room temperature with gentle shaking. For polymerization, we added 0.1% APS and 0.1% TEMED into monomer solution and placed the samples in a covered and humidified chamber, which was placed in an incubator at 45 °C for 1 hour.

Excess newly polymerized secondary gel was removed using a scalpel, and the gel-reembedded samples were washed three times (for 15 min each time) with 1X PBS. Before secondary gel expansion, typically we performed an additional DAPI staining step [DAPI (Thermo Fisher 62248) diluted 1:3000 in 1X PBS overnight at 4 °C] to facilitate localization of the iteratively expanded sample by the naked eye. The next day before mounting and imaging, the samples were subjected to four ddH$_2$O-based expansion steps (for 30 min each time) at room temperature with gentle shaking.

## Sample mounting

To mount the expanded samples embedded in the hydrogel, we used edge-cut round coverglasses to minimize the distance between the detection lens of our imaging system and the actual tissue sample. For the KA- and PKA-expanded samples, the gel could be attached to edge-cut 5-mm or 8-mm Ø coverglasses by pre-coating the coverglasses with 10-20 μL poly-L-lysine (PLL) at a concentration of 0.01% (Sigma P4707) and then allowing the coverglasses to dry at 37 °C for 20 min. These PLL-coated coverglasses could be stored at 4 °C for up to 1 month. For the relatively large re-KA- and re-PKA-expanded samples, normally reaching 1.5 ×2 x 1 cm, we used 3 M super glue gel to attach the gel to an edge-cut 15-mm Ø coverglass (Warner Ins. 64-0703-CS-15R). Mounted samples were allowed to stand for ~10 min for the glue to dry and then the coverglass containing the standing gel-embedded sample was glued to the imaging holder and placed into the imaging chamber, before filling the chamber with ddH$_2$O. All relevant chemical reagents are listed in Supplementary Table 8.

## Statistics and reproducibility

KA/re-KA or PKA/re-PKA expansion factors for each sample were determined by imaging whole specimens (cultured cells, spheroids, and fly brains) with a confocal microscope before and with our home-built lightsheet microscope (ΔBLX) after the expansion (Figs. 1, 3, 9 and Supplementary Fig. 15). For iterative expanded samples were imaged by ΔBLX and comparison of the size and shape in 3D rendering (Figs. 2, 4 and Supplementary Fig. 8). The expansion factor was determined by measuring the distance between two landmarks in the specimens in 2D or 3D. No statistical method was used to predetermine sample size.

## Reporting summary

Further information on research design is available in the Nature Portfolio Reporting Summary linked to this article.

## Data availability

Due to the data size limit, only the binning lightsheet data or partial raw image data generated in this study have been deposited in the BioStudies database[53] under accession code S-BIAD1488. Requests for the raw image data should be addressed to B.C.C. (chenb10@gate.sinica.edu.tw). Source data are provided with this paper.

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

## Acknowledgements

The authors would like to acknowledge financial support from the National Science Technology Council of Taiwan (NSTC112-2311-B-001-034-MY2 to T.Y.L., MOST 111-2124-M-001-003 to P.C., MOST 109-2628-M-001-001-MY4 to B.C.C.) and Academia Sinica of Taiwan (AS-IA-110-M04, AS-GCS-113-M01 to P.C., AS-TP-110-L10, AS-iMATE-110-43 to B.C.C.).

## Author contributions

X.T. and B.C.C. planned the expansion microscopy experiments. X.T. and M.J.T. performed the expansion microscopy experiments. B.C.C. performed lightsheet microscope construction and imaging acquisition. T.Y.L. and W.J.C. performed immunostaining experiments and fly preparation. P.T.L., H.C., C.M.L., C.H.T., J.C.H. and T.H.H. performed image processing and analysis. P.T.L., Y.C.T. and C.K.W. were responsible for AFM measurements. T.Y.L., Y.P.H., S.L., C.H.L., F.G.T., L.A.C., P.C. and B.C.C. were all involved in manuscript preparation. B.C.C. planned and managed the project.

## Competing interests

The authors declare no competing interests.
