## [Transparent Peer Review file · Nature Communications]

Rapid lightsheet fluorescence imaging of whole *Drosophila* brains at nanoscale resolution by potassium acrylate-based expansion microscopy

Corresponding Author: Dr Bi-Chang Chen

Version 0:

Reviewer comments:

Reviewer #1

(Remarks to the Author)

<What are the noteworthy results?>

The manuscript reports a novel expansion method that yields 30x-plus expansion of whole *Drosophila* brains and a new light sheet microscope that enables rapid imaging of such samples with limited photobleaching.

<Will the work be of significance to the field and related fields? How does it compare to the established literature? If the work is not original, please provide relevant references.>

The work may be of significance to the field and related fields. However, the primary problems the authors purport to resolve here have already been solved in the existing literature. Furthermore, the most outstanding results reported here are not reliably documented or have inconsistencies with the conclusions that generate reservations about the significance of the work at this stage.

Specifically, the authors claim that no expansion method exists or do not cite any existing method that enable thick tissue to be expanded beyond 4x within a robust gel. However, multiple published methods report this for tissues with similar or greater thickness than that of the *Drosophila* brain including an 8x method in Lillvis et al., 2022 (eLife), 10x method in Damstra et al., 2022 (eLife), 20x method in M'Saad et al., 2022 (bioRxiv), and a 16x method in Tavakoli et al., 2024 (bioRxiv). This does not make the method described here irrelevant by any means, but multiple robust-gel 8x-plus expansion methods for tissues of similar and greater thickness to that described here have been published by independent groups. Further, the notably high expansion factors reported here (30x-plus) that would be most novel and useful are achieved with non-robust gels (Sup. Table 4), making the advantage of this new method less apparent.

Furthermore, light sheet microscopes that enable comparable acquisition speeds and resolutions to the one described here also exist. For example, the widely available ZEISS LightSheet 7 enables effectively similar imaging capabilities to those described here.

Finally, use cases beyond those that were demonstrated here have already been reported using 4X and 8X whole brain *Drosophila* expansion protocols. Using 4x expansion, Gao et al., 2019 (Science) demonstrated the ability to segment individual neurons in sparsely and quasi-densely labeled samples and the ability to identify individual synapses. Using 8x expansion, Lillvis et al., 2022 (eLife) demonstrated the ability to segment individual neurons in sparse and quasi-densely labeled samples, to quantify synaptic connectivity between identified neurons and compare this connectivity to EM, and to quantify synaptic connectivity between identified neurons across individuals in relation to sex, neurophysiology, behavior, and experience. Using 4x expansion, Sanfilippo et al., 2024 (Neuron) demonstrated the ability to segment neurons in sparsely labeled samples and to characterize the molecular properties of synapses. The method described here may enable neuron segmentation in densely labeled samples (the above approaches do not allow this), but that isn't demonstrated in the manuscript. Furthermore, the ability to segment individual neurons in densely labeled samples has been demonstrated recently in mouse using a different approach that utilizes 16x expansion (Tavakoli et al., 2024 (bioRxiv)).

Thus, we already have the ability to generate robust 8x plus gels of *Drosophila* brains (and other comparably sized tissue), to image those samples rapidly with limited photo bleaching, and the ability to characterize molecular, synaptic, and anatomical properties of the neural circuits across individual animals. The work described here may provide another potential way to accomplish these things that may prove useful, but I would contend that the methodological limitations it purports to solve have already been solved in previously published, well-documented work.

<Does the work support the conclusions and claims, or is additional evidence needed?>

Additional evidence is needed for several of the author's claims.

Most importantly, the authors appear to be discussing single samples in multiple cases. For example, it seems they have one sample that expanded 32x. Is this repeatable? They discuss a single instance where the expansion factor was apparently lower than expected and explain it was likely due to the mounting medium used (page 8). Better documentation and evidence that they have a protocol that can reliably produce brains with 10x-plus expansion factors is required. Further, the particularly high expansion factor gels (30x plus) are apparently not strong (Sup. Table 4). This would seem to detract significantly from the impact of the method given several methods that produce strong gels in the 8-16x range already exist.

Along these lines, the reported 60x expansion factor discussed starting on page 11 is a bit confusing to me as well. Previous expansion factor calculations were based on whole brain size comparisons, but here the axon diameter is used. What is the size of the brain of this sample?

In a several instances the authors claim to visualize synaptic puncta (e.g., page 9), but they are not labeling synaptic proteins so cannot readily observe this. Presynaptic boutons, but not synaptic puncta, are clear in the images. They also speculate on visualizing mitochondria and other cellular organelles multiple times (e.g., page 9) without labeling them specifically. It may be the case, but I would remove such claims without independently labeling such structures.

The segmented neuron figures are a bit difficult to understand. It's difficult to directly relate the segmented neuron meshes to the fluorescent images. Some images/videos directly showing the fluorescent neurons with the translucent segmentation overlaid would be helpful. Perhaps notably, the authors claim that their method enables them to observe more complex profiles of TM5a than is revealed in the EM (page 11, line 25). An alternative interpretation is that they have not properly segmented the individual neurons here.

<Are there any flaws in the data analysis, interpretation and conclusions? Do these prohibit publication or require revision?>

See above.

<Is the methodology sound? Does the work meet the expected standards in your field?>

See above on the need for additional documentation on the ability to reliably generate reported expansion factors.

<Is there enough detail provided in the methods for the work to be reproduced?>

I cannot readily state whether there are enough details to build the described microscope, but there appear to be sufficient details on the fly genotypes and expansion protocols to reasonably attempt reproduction.

Reviewer #2

(Remarks to the Author)

In this manuscript, Tian and Lin et al. develop an iterative Potassium Acrylate (KA) and Potassium Polyacrylate (PKA) based expansion microscopy protocol in combination with volumetric imaging using a custom designed Bessel beam lightsheet microscope. The authors first demonstrate the stability of their KA hydrogel in preserving large samples in comparison to typically used Sodium Acrylate based hydrogels. They first use their method to image the whole brain from adult *Drosophila melanogaster* expressing transgenic fluorophore. Then they iteratively expanded the sample 1.5x to achieve a 13x expanded sample which could be imaged without considerable photo-bleaching. The authors further show the ability to reach a 60x expansion with PKA and re-PKA hydrogels that allow obtaining super-resolution images of a whole *Drosophila* brain. The authors also demonstrate the ability of their system to perform multicolored imaging and compare their images with images obtained from electron microscopy.

Overall, the technique developed in this manuscript is very interesting and can become relevant for large tissue imaging at subcellular resolutions. Additionally, the use of KA and PKA hydrogels to obtain hydrogels which are easier to handle is a notable improvement in the expansion microscopy field. However, there are a few minor issues with the text and associated figures which have been listed below. Overall, I recommend that the authors address the issues listed below before considering this manuscript for publication.

Major issues:

1. The authors show the quantification of expansion factor and isotropic expansion for KA-ExM and PKA-ExM. However, there is no quantification of expansion factor and isotropic expansion for re-KA-ExM and re-PKA-ExM in the manuscript. How isotropic is the re-KA-ExM and re-PKA-ExM?

2. The manuscript will benefit from quantitative evaluation or comparison of the different imaging modalities. For example, the authors can possibly evaluate the Full width at half maximum pre and post expansion to demonstrate the increased resolution after expanding the samples.

Minor issues in text:

1. On line 4 of page 7, it is mentioned that the sample was too large for homogeneous polymerization. Does this non-homogeneous polymerization cause an anisotropic expansion?
2. On line 23 of page 8, it is mentioned that long-term exposure to mounting adhesive causes a smaller expansion factor. Does it cause non-homogeneous expansion across z-direction?
3. Is there any specific reason for setting the digestion temperature at room temperature instead of at 37 degree? Can the authors demonstrate that the digestion is complete and the tissue is cleared?
4. Is there any specific reason for using ddH₂O instead of salt solution (or buffer) to dissolve and mix monomers?
5. Is there any specific reason for not degrading the template gel used for re-embedding for the re-KA-ExM and re-PKA-ExM?
6. To make it easy for the general readers of Nature Communications the authors might consider dividing the result section into shorter subsections with descriptive titles.

Comments on figure:

1. For supplementary figure 1g, it is not clear how the experiments and statistics were done. Is it 9 identical cells expanded in one KA-ExM gel? The number of technical replicates is not stated.
2. Please make sure that the structure of the DMAA/PKA polymer chain in Figure 3a is correct. Will there be repeated KA groups?
3. In supplementary figure 4, it seems like the stitching isn't great. Is it possible to replace them with better images? Also, the gel doesn't seem to have isotropic expansion.
4. In figure 4 b&c, it seems like the gel expands more in the direction of the length of the rectangle. The relative location of the orange nuclei is not consistent between b and c. Can the figure legend be re-written? It's not clear how b and c are related to panel a.
5. Is referencing Supplementary Fig. 1c in Page 6 line 2 correct?

Reviewer #3

(Remarks to the Author)

An important challenge in understanding how neurons act within circuits is resolving the components of neurites and synapses. Although this has been previously hampered by light microscopy's diffraction limit, expansion microscopy (ExM) has allowed scientists to resolve small structures with impressive resolution. Most ExM protocols involve a sodium acrylate-based hydrogel, which, although effective, can be watery and unstable, limiting the degree of expansion and the shape integrity of the samples. Tian, Lin et al., solve this issue by making two new types of hydrogel—one using potassium hydroxide and acrylic acid, and the other using commercially available PKA. The new gels have increased mechanical stability—the KA-based ExM can be sectioned using a vibrotome—and improved expansion ability—the PKA-based ExM can induce a 32-fold isotropic expansion.

Imaging an expanded sample can also provide distinct challenges, most notably, a decrease in fluorescence intensity and a reduced Z-range due to the limited operational distances of the objectives. Tian, Lin et al build a Bessel beam light sheet to solve this issue. With two longer working-distance objectives, they are able to image expanded samples with less difficulty. The group then applies KA/PKA-based ExM to the delicate neurites of the *Drosophila* optic lobe, imaging dopaminergic neurons and optic lobe neurons with nanometer resolution. The images are impressive and they are able to visualize gaps in the neurites that they believe correspond to mitochondria, and densities that correspond to synapses. This protocol is an exciting addition to the ExM repertoire, and will prove quite useful to many scientists, whether studying *Drosophila* or another model organism.

Although the images are quite striking and beautiful, the figures are a little bit crowded, and could use some rearranging to make the paper easier to read. Also, while the paper is experimentally sound, there are a few experiments that should be performed to provide more evidence for predictions made within the manuscript. That said, any limitations I found were rather minor, and I believe that the paper should be accepted with minor revisions.

Typos/text edits:

- TH-Gal4 (tyrosine hydroxylase) is not defined in the manuscript. Please add a sentence to suggest that you are labeling dopaminergic neurons in the main text.
- Line 12: There is an extra period in the sentence “determined the 3D expression pattern of transgene, TH-GAL4 in expanded fly brain..”
- Page 9, Line 18: “expaned samples” should be “expanded”
- Page 12, line 12-13: Change “where one segmented lamina monopolar neuron L3 (membrane GFP, shown in blue) among photoreceptors in red.” to “shown in blue) IS among photoreceptors in red.”
- Page 12: line 21: Change “involvelemt” to “involvement”.
- Figure 6: change “gree” to “green.” (line 5, page 29)
- Supplementary Figure 10: DAPI misspelled as “DPAl” (line 3)

- Page 13, line 16: change “routin” to “routine”
- Page 13, line 17: change “fluorescen” to “fluorescent”, “singals” to “signals”

Figure edits:

- It would be nice to have a better definition of “expansion factor”: i.e. Yellow scale indicates the actual size of the image divided by the expansion factor.
- Most of the figures are crowded and have too many overlapping components. Some figures would benefit from being cleaned up and put into insets:
 - Figure 1F: Put the smaller ellipsoid body in a box at the corner of the figure, not within the ellipsoid body itself.
 - Figure 3B: Put the 1x brain into a separate inset.
 - Figure 4: it is too challenging to tell the difference between the samples with the top right overlaid on top of the PKA-ExM. Would be better as an inset in the left corner.
- Please make sure that all the components of each figure are defined:
 - Supplementary figure 4: There is too little description within the figure legend: Please mention that the small insets are pre-expansion samples, and the larger images are post-expansion.
 - Figure 2A-C Legend: please write out that the green box in 2D is an inset of the dotted box at Figure 2A (also, the proportions of the two green rectangles do not seem to be the same size). Also, please define the yellow/blue boxes in Figures 2B/C in the legend as well.
 - Figure 2G-J: define what the white/blue dotted circles are in your figures.
- Supplementary figure S6: it would be nice to have a zoomed-in inset where you can better see the colocalization of the KA-ExM with the re-KA-ExM. Also, what is the reasoning why the brain halves are labeled in two different colors (cyan/magenta)?
- Supplementary Figure 10 is cited before Supplementary Figure 9. Please switch the order of the labeling.

Experimental edits:

- Supplementary figure S7: Would be nice to have a corresponding NCadherin stain to show medulla layers for reference to the reader.
- The authors suggest that the knots in the synaptic areas are filled by mitochondria. This would be very simple to test with a mitotracker stain + GFP colabel.
- Figure 5D: The reconstructed image doesn't give enough context. Please provide the original fluorescence image used to separate the neurites.
- Supplemental Figure 8: the authors argue that ExM provides a more impressive image quality than EM. Would be nice to have some sort of quantification that shows this, rather than an “improved” tracing. It would also be nice to have a table that compares SA-ExM to KA-ExM.

Version 1:

Reviewer comments:

Reviewer #1

(Remarks to the Author)

Thanks to the authors for addressing my comments. I have a few additional comments/suggestions based on the revisions.

1) If I understand the effective resolution correctly, the effective axial resolution achieved at ~40x is ~75nm. State of the art segmentation approaches used in EM would indicate that this is not sufficient resolution for densely labeled neuron segmentation and I would say this is not exactly a resolution that is similar to EM. Therefore, I would suggest the authors change the language in the abstract and conclusion of the manuscript that suggests the effective resolution is comparable to EM and will allow dense neuron segmentation. There are many positive things to say about the approach, but I don't think those conclusions are supported by the current methods advances.

2) Thanks for the new documentation of expansion factors in Sup. Figs 3-5 and clarity on this in the manuscript. Given the manuscript is heavily about Drosophila brain PKA-ExM and rePKA-ExM, the images shown in the rebuttal but not the manuscript seem helpful. Do data from these images make up part of numbers in Sup. Table 5? I don't think it's necessary to include the images from the rebuttal in the paper, but please add N to Sup. Table 5 to indicate the number of samples that went into these numbers. It would also be better to provide an actual average +/- SD instead of approximate given you have the data.

a. On this point, the authors state that it is difficult to obtain expansion ratios in very large samples because it requires high-res imaging to get these numbers. However, it should be possible to calculate expansion factors from the widefield images of the brains pre and post expansion. Or at least post expansion in relation to the average size of the Drosophila brain. It looks like these images exist for the samples discussed. If that's not the case, I wouldn't suggest new experiments to get this, but if you have the data I would use it to calculate expansion factor using a common metric instead of whole brain size in some instances, central complex size in others, axon diameter in others, etc.

3) Line 22, page 9 indicates blue arrow figure 3c indicates empty spaces in the neural fibers, but it's very difficult to see this in the figure. Further, I would suggest stating what the arrows are meant to indicate in the figure legends – not just in the main text.

4) I would suggest moving the new data introduced in the discussion to the main body of the manuscript.

5) I think it is the case that Gao et al., 2019 used a 4X, not an 8X, protocol and that Lillvis et al., 2022 used an iterative 8x protocol. This differs from what is indicated in Sup. Table 1.

Reviewer #2

(Remarks to the Author)

The authors have satisfactorily addressed all of my prior concerns.

Reviewer #3

(Remarks to the Author)

In this paper, Tian, Lin et al., improve the resolution of traditional expansion microscopy by making two types of hydrogels with increased mechanical stability to improve resolution of imaging samples such as whole-mount *Drosophila* brains. The gels produced by the authors not only offer improved resolution, but they also simplify the protocol for the end user. Although a few controls were missing from the initial draft, the authors have performed several experiments and provided adjustments to the paper that improve its quality. The use of spheroids has added rigor to the quantification of the expansion factor, and the use of Mitotracker more conclusively shows what the authors purported in the initial draft. Finally, the construction of a Supplemental Table comparing previous expansion microscopy systems to their own better shows how their work is an advancement compared to previous protocols in the field. I therefore suggest that the paper be accepted.

One small correction however: I spotted another typo in the figure legends (page 25, line 15: "circels" should be "circles")

Manuscript ID: NCOMMS-24-32341-T (NCOMMS-24-32341A)

Paper title: Rapid Lightsheet Fluorescence Imaging of Whole *Drosophila* Brains at Nanoscale Resolution by Potassium Acrylate-based Expansion Microscopy

Authors: Tian, X. and Lin, T.-Y. *et al.*

Reply point-by-point

Reviewer #1 (Remarks to the Author):

<What are the noteworthy results?>

The manuscript reports a novel expansion method that yields 30x-plus expansion of whole *Drosophila* brains and a new light sheet microscope that enables rapid imaging of such samples with limited photobleaching.

Reply: Thanks for the reviewer's summary of our work. To make the revised manuscript more solid, we made several improvements, including the spatial resolution and a more systematic study of the expansion protocol.

<Will the work be of significance to the field and related fields? How does it compare to the established literature? If the work is not original, please provide relevant references.>

The work may be of significance to the field and related fields. However, the primary problems the authors purport to resolve here have already been solved in the existing literature. Furthermore, the most outstanding results reported here are not reliably documented or have inconsistencies with the conclusions that generate reservations about the significance of the work at this stage. Specifically, the authors claim that no expansion method exists or do not cite any existing method that enable thick tissue to be expanded beyond 4x within a robust gel. However, multiple published methods report this for tissues with similar or greater thickness than that of the *Drosophila* brain including an 8x method in Lillvis et al., 2022 (eLife), 10x method in Damstra et al., 2022 (eLife), 20x method in M'Saad et al, 2022 (bioRxiv), and a 16x method in Tavakoli et al., 2024 (bioRxiv). This does not make the method described here irrelevant by any means, but multiple robust-gel 8x-plus expansion methods for tissues of similar and greater thickness to that described here have been published by independent groups. Further, the notably high expansion factors reported here (30x-plus) that would be most novel and usefu are achieved with non-robust gels (Sup. Table 4), making the advantage of this new method less apparent.

Reply: Thank the reviewer for pointing out the strengths and weaknesses of our manuscript. We highlighted the differences from the existing protocols in the introduction and summarized the existing literature in **Supplementary Table 1**. We also clarified the inconsistencies with more experimental results and rewrote protocols for better documentation. We have added the aforementioned research works as new references 6, 11, 14, 15, and 16. We have corrected the expansion ratio shown in Sup. Table 4 (now **Supplementary Table 5**), and new **Supplementary Figure 16** measured by using 100 μm spheroids. We did not use the fly brain for the measurement

of expansion ratio because the iteratively expanded size of a fly brain was already out of the detection range in our microscope ~ 2.2 cm, and that was the reason we cut the *re*-PKA ExM gel into half and performed the measurement separately. Another practical reason for not using the fly brain for the pre-and post-iterative expansion process is that tissue clearing is required to get the high resolution of whole fly brain imaging. The fly brain samples treated with clearing reagents such as CUBIC make the sample slightly swollen, and it becomes tricky to perform the expansion after tissue clearing, followed by an iterative expansion process for the calculation of the expansion ratio. The denominator in the post-/pre-expansion is very important, for example, on page 12 line 5 of the original manuscript, we claimed an expansion ratio of $\sim 60x$ based on a comparison of axon fiber diameter (expanded axon diameter of ~ 17 μm in Fig. 5c; unexpanded axon diameter of ~ 0.3 μm in the confocal image of Supplementary Fig. 9c). The denominator is 0.3 μm in confocal images (actually 2 image pixels), and the expansion ratio of $\sim 56x$ was obtained by 17 $\mu\text{m}/0.3$ μm . In our opinion, this may not be a good measurement based on the small structures with the variance of the denominator. We removed “60x” from the text in the revised manuscript but reported our measurements for these two samples.

To better estimate the expansion ratios, we utilized 100 μm spheroids to calculate the iterative expansion ratio based on KA- and PKA ExM as shown in **Supplementary Fig. 16**. We got the expansion ratio for 20x plus and 40x plus, respectively. We made another effort to report the central complex stained with a synaptic protein marker, Brp nc82, for the whole brain, as shown in new **Figure 7**. We reported a neuropil, fan-shaped body (FB), not measuring small structures, with a size of ~ 80 - 90 μm . After *re*-PKA ExM treatment, the expanded fly brain has an FB of ~ 3.5 mm. With this large ROI measurement, the expanded ratio is about 40x plus, which is consistent with the spheroid’s results. And the discrepancy in Fig. 4, the *re*-PKA ExM is measured at 32x, which is mainly from the **iterative imaging pipeline** on the identical DAPI stained fly brain as shown in **Supplementary Fig. 7a**. The original fly brain was examined by epi-fluorescence microscopy. Following PKA ExM, the hydrogels were mounted and transferred to the ΔBLX chamber for 3D whole-brain imaging. After lightsheet imaging, the hydrogel was removed from the water chamber and subjected to *re*-PKA ExM. The *re*-PKA hydrogel was placed back into the ΔBLX chamber and underwent lightsheet whole brain imaging for characterization. The gel may suffer from shrinkage because of the interruption in the imaging and sample-transferring process, resulting in a lower expansion ratio for iterative measurement. **Fig5a**, **Fig. 6a**, and **Fig.7c** were conducted with *re*-PKA ExM end-products, with no sample transferring back and forth, resulting in greater expansion ratios.

Hopefully, we can answer the reviewer’s concern with these explanations and additional experimental results.

Furthermore, light sheet microscopes that enable comparable acquisition speeds and resolutions to the one described here also exist. For example, the widely available ZEISS LightSheet 7 enables effectively similar imaging capabilities to those described here.

Reply: For a commercial microscope like Zeiss Lightsheet 7, it is an excellent modality to image the expanded samples as well. However, several issues prevent such a microscope from imaging the extremely expanded samples with the size of 2 cm x 1 cm x 1 cm, shown in our manuscript at the pixel resolution of 300 nm. These issues are related to the following parameters in

the commercial microscope: (1) traveling range of the stage, (2) sample chamber size, and the objective lens.

In the Zeiss lightsheet system, the imaging range is (+-) 5 mm in x and z, with a sample mounted “accurately” with its x-z center to the rotation axis, which translates the 1 cm imaging length in x and z. Note that there is still a certain amount of the embedded gel surrounding the expanded tissues, which makes the whole gel sample even larger than 1 cm. However, the main problem is the sample chamber size. The chamber for water imaging is too small in volume to comfortably image the aforementioned-sized block. There is not much space to make a customer-designed version of this water chamber to enlarge the space inside the chamber due to the optical design of the current Zeiss lightsheet system. Moreover, the objective lens is a 10x / 0.5 NA water dipping objective having a working distance of 3.7 mm, which makes it impossible to image such thick samples presented in our manuscript. Last but not least is the laser power. The laser power provided by Zeiss Z7 at the individual wavelengths couldn’t give a good contrast for such diluted fluorescence signals due to the high expansion ratio of the samples at a reasonable exposure time, such as 100 ms/frame. Based on all these concerns, we have constructed our Bessel lightsheet microscope features with two extremely long working distance at NA of 0.6 (excitation: 15 mm, detection: 8mm) with an imaging pixel resolution of 325 nm and a customized laser combiner system with high output laser power, which could provide good temporal resolution at 100 ms/frame in our cases.

Finally, use cases beyond those that were demonstrated here have already been reported using 4X and 8X whole brain *Drosophila* expansion protocols. Using 4x expansion, Gao et al., 2019 (Science) demonstrated the ability to segment individual neurons in sparsely and quasi-densely labeled samples and the ability to identify individual synapses. Using 8x expansion, Lillvis et al., 2022 (eLife) demonstrated the ability to segment individual neurons in sparse and quasi-densely labeled samples, to quantify synaptic connectivity between identified neurons and compare this connectivity to EM, and to quantify synaptic connectivity between identified neurons across individuals in relation to sex, neurophysiology, behavior, and experience. Using 4x expansion, Sanfilippo et al., 2024 (Neuron) demonstrated the ability to segment neurons in sparsely labeled samples and to characterize the molecular properties of synapses. The method described here may enable neuron segmentation in densely labeled samples (the above approaches do not allow this), but that isn’t demonstrated in the manuscript. Furthermore, the ability to segment individual neurons in densely labeled samples has been demonstrated recently in mouse using a different approach that utilizes 16x expansion (Tavakoli et al., 2024 (bioRxiv)).

Reply: We thank the reviewer’s comments on these excellent works. As the reviewer mentioned previously, there are still a lot of efforts to increase the expansion ratio, as shown in these aforementioned bioRxives reports, showing the demand for an even higher expansion ratio for various applications. Moreover, there are still two orders of magnitude differences between optical microscopy and electron microscopy. In our opinion, expansion microscopy is probably a good approach to bridge this gap by sample preparation with a high expansion ratio. Therefore, given that the fluorescence intensity is maintained, a 100x expansion ratio is probably the goal we would like to achieve. That’s why we present the work with extremely expanded samples and build an imaging platform- an axicon-based lightsheet microscope to image such a whole largely

expanded gel. On page 3 line 20, “In addition, whole-brain scale mapping of the neural architecture at higher spatial resolution may be further boosted by increasing the sample expansion ratio.” On page 16 Line 19 “Alternatively, in order to meticulously delineate the intricacies of dendritic processes or synaptic structures with whole brain coverage, a greater expansion ratio is required, resulting in rather bulky samples.” On page 18 line 6 “By means of a greater sample expansion ratio from iterative process and axicon-based Bessel lightsheet, imaging the nanoscopic molecular architecture of features such as the synaptic active zone, T-bar shown in the EM of Supplementary Fig. 17b, within the whole brain (as exemplified in Fig. 7 and Supplementary Fig. 14b) is now possible.”

Of course, all the works herein are based on previous excellent works that inspire us to explore a more technology-driven orientation. Alternatively, we are providing different gel chemistry to fit expanded gel into our lightsheet platform, which potentially will be a good option for other applications. We have spent more effort to show more densely labeled neuron segmentations shown in the revised **Figure 5b**, new **Supplementary Figure 12**, and **Supplementary Movie 5**, as shown above, the top-view volume rendering and side-view. Apparently, with the higher sample expansion ratio, a lower signal-to-noise ratio is expected, making the 3D imaging segmentation harder. Currently, we are trying to use the segmentation data we obtained as the training library to speed up the 3D imaging segmentation using the AI approach. Hopefully, in the near future, we could present better and more densely labeled samples. With the new segmentation results shown in the revised **Figure 5b**, new **Supplementary Figure 12**, and **Supplementary Movie 5**, we hope the reviewer will appreciate our effort to realize the density segmentation. In Tavakoli et al., 2024 (bioRxiv), the mouse brain thin slice sample ($396 \times 109 \times 22 \text{ } \mu\text{m}^3$), as summarized in Supplementary Table 1, the sample is relatively thin, with a thickness of $\sim 350 \text{ } \mu\text{m}$ after expansion at the 16x. Moreover, the way they performed the iterative process was to crop the small region of interest from 1st gel and conduct the second expansion, which made the protocol and imaging easier but with a limited view. The data we present here is for the whole fly brain at more than 30x expansion and 3D imaging of several millimeters thick. The iterative protocol could also be

applied to whole fly brains. Note that the expansion ratio affects volume and signals cubically, $(\sim 32/16)^3 = \sim 8$, and the thickness of the sample becomes \sim half a centimeter in our cases. Different imaging modalities should be developed to image such a large and thick sample, as shown in our manuscript.

Thus, we already have the ability to generate robust 8x plus gels of *Drosophila* brains (and other comparably sized tissue), to image those samples rapidly with limited photo bleaching, and the ability to characterize molecular, synaptic, and anatomical properties of the neural circuits across individual animals. The work described here may provide another potential way to accomplish these things that may prove useful, but I would contend that the methodological limitations it purports to solve have already been solved in previously published, well-documented work.

Reply: Thanks for the reviewer's comments on our work. As the reviewer mentioned previously, there are still a lot of efforts on increasing the expansion ratio as exemplified in these bioRxives reports such as 16~20x. Why do we stop here? As one of the technical developers, we would like to push the expansion ratio further to fill the gap between optical microscopy and electron microscopy; moreover, since expansion microscopy is a potential candidate to realize such a beautiful work, especially for the whole fly brain imaging community, the motivation for this present research is to image the whole largely expanded fly brains by expansion lightsheet microscope. Even though there is already related research about expansion lightsheet microscopy demonstrated previously, 4x or 8x fly brains. We are thinking about how to image the expanded brains with 20x plus or even more as a whole without physical cutting. To respond to this issue, the manuscript presents another expansion protocol (potassium acrylate-based ExM) that could easily fit the designed two long-working distance objectives housed in lightsheet microscope (Δ BLX) by direct sample scanning with free-standing hydrogel. The expanded hydrogel with good mechanical strength could be moved during sample tiling for imaging stitching and analysis. The pipeline of acrylate-based ExM lightsheet could facilitate comparative analyses with N number in a fast fashion, such as comparisons between wild-type and mutant counterparts, which are beyond the reach of volume-based EM connectome reconstructions. We have added new **Figure 7**, demonstrating the spatial resolution that could be obtained in brain-wide regions by labeling synaptic protein, Brp (as shown below), the scaffold clusters (arrowheads), a major and essential constituent of T-bars (~ 140 nm). These figures display different orientations of Brp scaffold proteins and are either separate or grouped. Moreover, color-coding of the Brp clusters in **Fig. 7f** could be mapped optically to a depth of 30 μ m due to the highly expanded nature of the Brp scaffold, as shown in the inset of **Fig. 7f**. It becomes possible to map the orientation of the scaffold protein (Brp) as shown in the inset of (c) within the whole brain without physical sectioning for the selected region. On page 14,

"As further evidence for the excellent spatial resolution gained from potassium (poly)acrylate-based ExM and lightsheet microscopy, we imaged the molecular architecture of synaptic active zones based on the nanoscopic distribution of the scaffold protein Brp labeled by a monoclonal antibody (Brp^{Nc82}) in the adult fly brain.^{11, 36} The central complex and accessory neuropils are shown in **Fig. 7c**, with a fan-shaped body (FB) of ~ 3.5 mm marked in yellow. Note that the original size of the FB is about 80~90 μ m³⁷, revealing a ~ 50 x plus expansion ratio through *re*-PKA-ExM. 3D volume rendering of the asymmetrical body (AB) is also shown in blue in **Fig. 7c**³⁸, constructed by

2000 x 2600 x 301 pixels at a pixel resolution of 325 nm x 325 nm x 5 μ m selected from the whole-brain image. An enlarged MIP view of the C-terminal epitope (Brp^{Nc82}) in combination with Cy3 labeling is presented in **Fig. 7d**, and a series of slice views at a z interval of 5 μ m is shown in **Supplementary Fig. 14b**. Note that the boundary for intensity variation is caused by image stitching. We segmented ~261 Brp clusters in the upper left part of the image (highlighted by surface rendering). Magnifications of the identified active zones (Brp clusters) are displayed in **Fig. 7e** and **7f**. Note that the scaffold clusters (arrowheads), a major and essential constituent of T-bars, display different orientations and are either separate or grouped. Moreover, color-coding of the Brp clusters in **Fig. 7f** could be mapped optically to a depth of 30 μ m due to the highly expanded nature of the Brp scaffold, as shown in the inset of **Fig. 7f**."

<Does the work support the conclusions and claims, or is additional evidence needed?>

Additional evidence is needed for several of the author's claims.

Most importantly, the authors appear to be discussing single samples in multiple cases. For example, it seems they have one sample that expanded 32x. Is this repeatable? They discuss a single instance where the expansion factor was apparently lower than expected and explain it was likely due to the mounting medium used (page 8). Better documentation and evidence that they have a protocol that can reliably produce brains with 10x-plus expansion factors is required. Further, the particularly high expansion factor gels (30x plus) are apparently not strong (Sup. Table 4). This would seem to detract significantly from the impact of the method given several methods that produce strong gels in the 8-16x range already exist.

Reply: We thank the reviewer's suggestions. Sorry for the confusion about the measurements. We prepared a new **Supplementary Figure 7** to further clarify the ideas we carried out to determine the expansion ratio. To fully understand the iterative process in terms of structure deformation for potassium acrylate (KA) or potassium polyacrylate (PKA) ExM, we set up an imaging pipeline, as shown below. The original fly brain was examined and measured using epifluorescence microscopy. Following KA/PKA ExM, the hydrogels were mounted and transferred to the Δ BLX chamber for 3D whole-brain imaging. After lightsheet imaging, the hydrogel was removed from the water chamber and subjected to *re*-KA/PKA ExM. The *re*-KA/PKA hydrogel was placed back into the Δ BLX chamber and underwent lightsheet whole brain imaging for characterization. The fly brain was treated with PKA-ExM and expanded to 12x after lightsheet imaging of the whole brain as shown in **Fig. 4a** top right. We transferred this 12x PKA gel back to the bench for a flowing iterative process (*re*-PKA); the 2nd hydro gel was put back into the chamber, and whole brain imaging was performed. However, we found this *re*-PKA gel was larger than 2 cm, which was beyond the imaging detection field of view due to the traveling range of the stage

(22mm). Hence, we cut this 2nd gel in half and performed the lightsheet imaging separately, as shown in the new **Supplementary Fig.8** (cyan and pink color). We measured the original fly brain size and *re*-PKA ExM size, we got an expansion ratio of 32x. But in our mind, we would expect the expansion ratio even higher (> 50) since the iterative process was a 4x protocol. The PKA-gel became smaller after being imaged in the lightsheet imaging chamber. The reason may be attributed to the way we mount the samples with 3M glue since PKA gel is sensitive to the chemicals. Due to this shrinkage, we end up with a smaller expansion ratio (~32x in this case) for the *re*-PKA ExM by iterative imaging pipeline, shown below.

To address this problem, we only conducted the lightsheet imaging for the end-product of *re*-PKA, as shown below. By doing this, we found the *re*-PKA gel became larger than the cases shown in **Figures 5, 6, and new Fig. 7c**. Again, these *re*-PKA gels exceeded the detection limit due to the gel size. We could use either an optical lobe or a central complex for imaging. With the size calculation, we got the expansion ratio of > 40. This expansion ratio was obtained with multiple samples throughout the manuscript.

One more piece of evidence, we performed a smaller biological sample to measure the whole sample pre- and post-iterative PKA process. We used 100 μm spheroids for testing, and we measured five trials each for the *re*-KA and *re*-PKA processes. We ended up with the ~20 and ~40 expansion ratios, as shown in **Supplementary Figure 16** and the figure below. “The expansion ratio of *re*-KA and *re*-PKA ExM based on 100 μm spheroids generated from the HCT116 cell line in which nuclei have been stained by DAPI and imaged by ΔBLX . (a) Five trials of *re*-KA ExM treatment, resulting in a ~20x-plus expansion ratio. (b) Five trials of *re*-PKA ExM treatment, resulting in a ~40x-plus expansion ratio. (c) Image of the original ~100- μm spheroid. (d) Lightsheet imaging of the spheroid shown in (c) after the *re*-PKA-ExM process, resulting in a ~40x-plus expansion to ~4 mm (the original size is shown in the inset at the same scale). (e) Raw 3D MIP image of one ~300- μm mitotic cell selected from (d) after *re*-PKA-ExM. (f) Single slice view of (e) showing the detailed structure of the chromosomes. (g) Raw 3D MIP image of one 330- μm interphase nucleus selected from (d) after *re*-PKA-ExM. (h) Single slice view of (g) showing the detailed structure of the nucleus.”

Regarding the repeatability of the *re*-PKA ExM, we have performed a batch of 5 fly brains for the *re*-PKA process. The results are shown below, where #3-fly brain failed in the gelation process. But others are doing well.

In order to make the protocol concise and clear to follow, we have restricted and rewritten the methods section of the protocols as shown in the new manuscript (see underlined text) to improve the documentation. All the buffer components are included in the **new Supplementary Table 7** (as shown blown).

Potassium acrylate (KA) or potassium polyacrylate (PKA) expansion microscopy

The anchoring treatments were performed in 10 mM MA-NHS overnight at 4 °C, and washed twice with 1X PBS for 5 min prior to a gelation step the next day. KA monomer stock was prepared according to one-step in situ synthesis. To make a total of 60 mL KA monomer stock, 11.2 g KOH was dissolved in 36 mL ddH₂O, before adding 14.2 mL of acrylic acid (AA-liquid) into the completely dissolved KOH. When the KOH and AA were well mixed after stirring for 30 min, 17 mg MBA was added into the mixture and the total volume was adjusted to 60 mL. The resulting monomer solution was aliquoted for convenient use (450 μL per tube) and frozen immediately at -20 °C to maintain the monomer's freshness. PKA monomer stock was prepared by first dissolving 1.31 g PKA in 10 mL ddH₂O, and then adding the mixture into an entire 5-mL pack of DMAA liquid, before adjusting the final volume to 16 mL with ddH₂O. The monomer solution was aliquoted (450 μL per tube) and frozen at -20 °C immediately. To prepare ready-to-use gelling solution, a 450-μL aliquot of KA or PKA monomer solution was vacuumed for 10 min, de-vacuumed with N₂ gas for 10 min, and this cycle was repeated once more. After placing a sample onto a 10-mm coverglass in a 3-cm petri-dish cushioned with a flat piece of parafilm, the gelling solution was mixed by adding 50 μL of stock KPS solution (0.036 g/mL in ddH₂O) and 2 μL of 100% TEMED into the de-gassed 450 μL monomer right before use. The ready-to-use gelling solution (30 μl) was then added to the sample and covered with another 10-mm coverglass to make a sandwich-like gelation chamber. The 3-cm petri-dish containing the sample was then transferred into a covered humidified chamber, and gelation was performed at 37 °C for 2 hours. After sample gelation, the coverglasses were detached and excess gel was removed using a scalpel. The gel-embedded

sample was then transferred to a small container with digestion buffer. Digestion was performed overnight at room temperature with gentle shaking.

The next day, digested KA gels were transferred into 6-cm petri-dishes and washed three times with 1X PBS for 15 min each time. Gels were stained using DAPI in 1X PBS (1:3000, ThermoFisher 62248) for 4 hours at room temperature, and transferred into new 6-cm dishes three times (30 min each time) for water expansion, followed by overnight shaking in water at room temperature. Digested PKA gels were transferred into 6-cm petri-dishes for expansion in water for 10 min (once), then with DAPI (1:500 diluted in ddH₂O, SouthernBiotech 0100-20) for 4 hours (once), with a final water wash overnight at room temperature with gentle shaking. The samples were expected to reach maximum extent by the following morning and could then be mounted for imaging by lightsheet microscopy.

re-KA and re-PKA expansion microscopy

The iterative re-KA and re-PKA expansions began with the KA-expanded or PKA-expanded brains, respectively. The re-embedding and secondary gel processes were the same for re-KA and re-PKA expansion, unless otherwise specified. First, excess gel was removed using a scalpel, making sure to leave as little surrounding gel as possible. Typically, the KA-expanded *Drosophila* brain reached 5 x 3 x 2 mm in size, so the gel was cut to 6 x 4 x 3 mm using DAPI signal under a dissection microscope as a guide to sample location in the gel and for incisions. The resulting gel-embedded sample was washed twice for 10 min at room temperature in re-embedding wash buffer prepared freshly for each experiment, and then once again in the same re-embedding wash buffer for 1 hour. Then, the re-embedding gelling solution was prepared. The subsequent gelation procedure was similar to that described above for KA-ExM gels, using two coverglasses (12-mm Ø, Warner Ins. 64-0702-CS-12R) to make a sandwich chamber for gelation. Re-embedding gelation occurred at 45 °C for 2 hours.

After re-embedding gel polymerization, excess gel was again removed using a scalpel, and the gels were washed three times with 1X PBS for 15 min each time. For gel polymerization in advance of the second expansion, samples were first washed twice (10 min each time) with the second gelling solution, and then once with the same gelling solution for 1 hour at room temperature with gentle shaking. For polymerization, we added 0.1% APS and 0.1% TEMED into monomer solution and placed the samples in a covered and humidified chamber, which was placed in an incubator at 45 °C for 1 hour.

Excess newly polymerized secondary gel was removed using a scalpel, and the gel-reembedded samples were washed three times (for 15 min each time) with 1X PBS. Before secondary gel expansion, typically we performed an additional DAPI staining step [DAPI (ThermoFisher 62248) diluted 1:3000 in 1X PBS overnight at 4 °C] to facilitate localization of the iteratively expanded sample by the naked eye. The next day before mounting and imaging, the samples were subjected to four ddH₂O-based expansion steps (for 30 min each time) at room temperature with gentle shaking.

Supplementary Table 7. Buffer components and chemical name abbreviations.

BUFFER NAME	COMPONENTS	STORAGE CONDITION	USE CONDITIONS
Fixation solution	4% Paraformaldehyde (PFA), 0.2% Glutaldehyde (GA)	4 ° C	40 min in 4% PFA, then 20 min in 0.2% GA, 4 ° C
Blocking buffer	10% Normal goat serum (NGS), 2% Triton X-100, 0.02% Sodium azide in 1X PBS	4 ° C	1hr, 37 ° C
Antibody dilution buffer	1% NGS, 0.25% Triton X-100, 0.02% Sodium azide in 1X PBS	4 ° C	1-3 days, RT
Antibody wash buffer	1% Triton X-100 in 1X PBS	4 ° C	15 min, x3 times, RT
Anchoring solution	10mM Methacrylic acid N-hydroxy succinimidyl ester (MA-NHS) in 1X PBS	4 ° C	Overnight, 4 ° C
KA monomer solution	18.7% (w/v) Potassium hydroxide (KOH), 23.7% (v/v) Acrylic acid (AA-liquid), 0.03% (w/v)N,N'-methylenebisacrylamide (MBA) in ddH ₂ O	-20 ° C	For gelation, 2hrs, 37 ° C
PKA monomer solution	21.8% (w/v) Potassium polyacrylate (PKA), 31.3% (v/v) Dimethylacrylamide (DMAA), in ddH ₂ O	-20 ° C	For gelation, 2hrs, 37 ° C
Gelation initiator	0.0036 g/mL Potassium peroxydisulfate (KPS), 0.4% Tetramethylethylenediamine (TEMED)	-20 ° C for KPS, RT for TEMED	For gelation, 2hrs, 37 ° C
Digestion buffer	50 mM Tris pH 8.0, 800 mM guanidine HCl, 2 mM CaCl ₂ , and 0.5% (v/v) Triton X-100 in ddH ₂ O, with freshly added proteinase K diluted 1:100	-20 ° C for Proteinase K, RT for others	Overnight, RT
Re-embedding wash buffer	13.75% Acrylamide (AA), 0.038% MBA, 0.03% ammonium persulfate (APS), 0.03% TEMED, and 5 mM Tris pH 8.0 in ddH ₂ O	Freshly prepared	10 min, x2 times, then 1 hr, RT
Re-embedding gelling solution	13.75% AA, 0.038% MBA, 0.05% APS, 0.05% TEMED, and 5 mM Tris pH 8.0 in ddH ₂ O	Freshly prepared	2 hrs, 45 ° C
Second gelling solution	2 M NaCl, 8.6% (w/w) sodium acrylate (SA), 2.5% (w/w) AA, 0.15% (w/w) MBA in 1X PBS, with freshly added 0.03% APS and 0.03% TEMED	Freshly prepared	1 hr, 45 ° C

Regarding the issue for Supplementary Table 4 (new **Supplementary Table 5**), we added the measurements of identical ones for pre- and post-expansion from spheroids data and fly brains to support the expansion ratio we obtained. It is quite challenging to measure the pre- and post-expansion for a highly expanded ratio (>40x) at the same pixel size resolution. But we do show the possibility here with the present lightsheet and expansion protocol, and hopefully, there will be more data sets involved in the future with the protocols open to the community.

Along these lines, the reported 60x expansion factor discussed starting on page 11 is a bit confusing to me as well. Previous expansion factor calculations were based on whole brain size comparisons, but here the axon diameter is used. What is the size of the brain of this sample?

Reply: To show the measurement for pre- and post-expansion on the fly brain following the pipeline shown in **Supplementary Figure 7b** for end-product imaging. The *re*-PKA ExM fly brain has a size > 2cm, as shown in the new **Supplementary Figure 5b**, which is out of our size detection limitation due to the sample stage travelling range used in the lightsheet microscope. Therefore, we cut this ~2 cm fly brain with only one expanded optical lobe, which is measured and shown in **Figure 5**. And for the pre-expansion measurement, we chose similar structures for the size reference. As explained previously, in comparison of axon fiber diameter (expanded axon diameter of ~17 μm in **Figure 5c**; unexpanded axon diameter of ~0.3 μm in the confocal image of **Supplementary Figure 9c**) the denominator is 0.3 μm in confocal images (actually 2 image pixels), and a ~ 56x expansion ratio is obtained by 17/0.3 μm . In our opinion, this may not be a good measurement based on the small structures for the variance of the denominator. We removed “60x” from the manuscript but kept the measurements for these two samples as the references.

In a several instances the authors claim to visualize synaptic puncta (e.g., page 9), but they are not labeling synaptic proteins so cannot readily observe this. Presynaptic boutons, but not synaptic puncta, are clear in the images. They also speculate on visualizing mitochondria and other cellular organelles multiple times (e.g., page 9) without labeling them specifically. It may be the case, but I would remove such claims without independently labeling such structures.

Reply: We thank the reviewer’s suggestions. We have changed our terms to “presynaptic boutons”, as shown on page 9, with the green frame displaying punctate pre-synaptic structures (indicated by a white arrow) of similar size to those presented in Fig. 2j. and page 10, The sub-volume framed in yellow in Fig. 3d shows the enhanced resolution of pre-synaptic boutons (white arrow), enabling better visualization of cellular organelles (empty spaces likely representing mitochondria) inside the pre-synaptic boutons compared to the images shown in Fig. 2j.

To validate the speculated mitochondria structures based on the optical images in **Figure 2j, 3d** by expansion lightsheet imaging and **Supplementary Fig. 6c** from EM data, we generated a fly strain, UAS-mCherry.mito.OMM/+; TH-GAL4, 20XUAS-6XGFP/+ fly to specifically label the mitochondria and performed two-color PKA-ExM lightsheet imaging. The related data is present in the new **Figure 7a** and **b**, which support our finding that the empty space is occupied by mitochondria. (a) Two-color MIP image of GFP-labeled TH-GAL4 (green) and mCherry-labeled

mitochondria (red) in the PRW region of the PKA-ExM fly brain. (b) Expanded slice view of the red box in (a) showing only mitochondria. (as shown below)

The segmented neuron figures are a bit difficult to understand. It's difficult to directly relate the segmented neuron meshes to the fluorescent images. Some images/videos directly showing the fluorescent neurons with the translucent segmentation overlaid would be helpful. Perhaps notably, the authors claim that their method enables them to observe more complex profiles of Tm5a than is revealed in the EM (page 11, line 25). An alternative interpretation is that they have not properly segmented the individual neurons here.

Reply: We agree with the reviewer's concern about the segmentation issue. Image segmentation is very important and becomes harder and harder while the expansion ratio increases since the single-to-noise ratio worsens. Related to **Figure 5d**, we have made a movie as a **new Supplementary Movie 4** with raw data overlaid with the segmentation to help the reader understand where the segmentation images come from and confirm that the complex profiles of Tm5a are not from the segmentation inaccuracy. Moreover, we have conducted a group of densely labelled neuron segmentations shown in the new inset of **Figure 5b**, **Supplementary Figure 12** and **Supplementary Movie 5**. Apparently, with the higher sample expansion ratio, the lower signal-to-noise ratio is expected, making the 3D imaging segmentation harder. Currently, we are trying to use the segmentation data we obtained as the training library to speed up the 3D imaging segmentation using the AI approach. Hopefully, in the near future, we could present better and more densely labeled samples. With the new segmentation results shown in the revised **Figure 5b**, new **Supplementary Figure 12**, and **Supplementary Movie 5**, and we also tone down the description on page 12 "Indeed, the optical images presented in Fig. 5d reveal comparable profiles as in the EM-imaged Tm5a neuron shown in Supplementary Fig. 11a." Since the EM data from the library probably not representative, there is still individual discrepancy here. It is not fair to base one EM image to reach a conclusion. Moreover, from the Tm5a neurons, we obtained in the dense area shown in the inset of **Figure 5b** and **Supplementary Figure 12**. We do observe the diversity of the dendritic processes (some complicated while some in simple profiles) as shown in the skeletonized 15 Tm5a neurons in the *re*-PKA ExM fly optical lobe (a) and (b) The

top view and side view of a group of skeletonized Tm5a neurons in M3, M6, and M8 layers. (c) The top view (left) and side view (right) of the individual and colored Tm5a neurons with dendritic processes in the M6 layer are marked with a white arrowhead in (b).

See above.

See above on the need for additional documentation on the ability to reliably generate reported expansion factors.

I cannot readily state whether there are enough details to build the described microscope, but there appear to be sufficient details on the fly genotypes and expansion protocols to reasonably attempt reproduction.

Reply: Many thanks for all the comments and suggestions made by the reviewer. We hope that our responses and new experimental data can answer the reviewer's concerns. For the axicon-based lightsheet microscope (Δ BLX) featuring the long working distances objective and voice-coil stage, all the equipment could be reached by the provided companies and compatible with lattice lightsheet microscopy. With the revised protocols in the method part, we believe the KA-ExM and PKA-ExM could be easily adapted by the readers to the other biological samples.

Reviewer #2 (Remarks to the Author):

In this manuscript, Tian and Lin et al. develop an iterative Potassium Acrylate (KA) and Potassium Polyacrylate (PKA) based expansion microscopy protocol in combination with volumetric imaging using a custom designed Bessel beam lightsheet microscope. The authors first demonstrate the stability of their KA hydrogel in preserving large samples in comparison to typically used Sodium Acrylate based hydrogels. They first use their method to image the whole brain from adult *Drosophila melanogaster* expressing transgenic fluorophore. Then they iteratively expanded the sample 1.5x to achieve a 13x expanded sample which could be imaged without considerable photo-bleaching. The authors further show the ability to reach a 60x expansion with PKA and re-PKA hydrogels that allow obtaining super-resolution images of a whole *Drosophila* brain. The authors also demonstrate the ability of their system to perform multicolored imaging and compare their images with images obtained from electron microscopy.

Overall, the technique developed in this manuscript is very interesting and can become relevant for large tissue imaging at subcellular resolutions. Additionally, the use of KA and PKA hydrogels to obtain hydrogels which are easier to handle is a notable improvement in the expansion microscopy field. However, there are a few minor issues with the text and associated figures which have been listed below. Overall, I recommend that the authors address the issues listed below before considering this manuscript for publication.

Reply: We thank the reviewer's comments on our work and highlight the strengths of our manuscript. Hopefully, with the revised manuscript, we could make our manuscript more solid and easy to understand for readers interested in the techniques. We modified our manuscript and conducted additional experiments to address the reviewer's comments. Please look into the following point-by-point reply.

Major issues:

1. The authors show the quantification of expansion factor and isotropic expansion for KA-ExM and PKA-ExM. However, there is no quantification of expansion factor and isotropic expansion for re-KA-ExM and re-PKA-ExM in the manuscript. How isotropic is the re-KA-ExM and re-PKA-ExM?

Reply: Thanks for the reviewer pointing this out. Since the *re-KA-ExM* and *re-PKA-ExM* fly brains have an expanded whole brain size exceeding 2 cm size easily, particularly for *re-PKA-ExM*, we have to cut the fly brain in half for the measurement as shown in **Figure 4** and **Supplementary Figure 8**. To show the systematic quantification of the expansion factor and broaden the sample diversity, we used the spheroid for the measurements and added the new **supplementary Figure 16**. The expansion ratio of re-KA and re-PKA ExM is based on 100 μm spheroids generated from the HCT116 cell line in which nuclei have been stained by DAPI and imaged by ΔBLX . (a) Five trials of re-KA ExM treatment, resulting in a $\sim 20\text{x}$ -plus expansion ratio. (b) Five trials of re-PKA ExM treatment, resulting in a $\sim 40\text{x}$ -plus expansion ratio. (c) Image of the original $\sim 100\text{-}\mu\text{m}$ spheroid. (d) Lightsheet imaging of the spheroid shown in (c) after the re-PKA-ExM process, resulting in a $\sim 40\text{x}$ -plus expansion to $\sim 4\text{ mm}$ (the original size is shown in the inset at the same scale).

2. The manuscript will benefit from quantitative evaluation or comparison of the different imaging modalities. For example, the authors can possibly evaluate the Full width at half maximum pre and post expansion to demonstrate the increased resolution after expanding the samples.

Reply: Thanks for the reviewer’s suggestion. We have added the new **Supplementary Figure 4** to characterize in situ KA-ExM (8x) and re-KA-ExM (13x) at high spatial resolution of the TH-GAL4, 20XUAS-6XGFP/+ fly brain as the following. (a) Line profile along a neuron fiber in a similar area of the EB. The full widths at half maximum measurements for the (cleared) original, in situ KA-ExM, and re-KA-ExM samples are 1.97, 7.38, and 16.85 pixels, respectively. (b) Line profile along two presynaptic boutons in the PRW region. The full width at half maximum measurements for the (cleared) original, in situ KA-ExM, and re-KA-ExM samples are shown. Note that the original image was acquired from a cleared fly brain in order to obtain a high resolution. The fly brains are already slightly swollen due to being subjected to the clearing reagent (CUBIC).

Minor issues in text:

1. On line 4 of page 7, it is mentioned that the sample was too large for homogeneous polymerization. Does this non-homogeneous polymerization cause an anisotropic expansion?

Reply: We thank the reviewer's question. The second expansion process was still isotropic. Therefore, the resulting *re*-KA and *re*-PKA samples were structurally maintained, but the expansion ratio did not reach 4x. This was possibly due to gel shrinkage (~2x shrinkage) during several rounds of washes in both re-embedding and 2nd gelation processes. In the revised manuscript, we removed this sentence to avoid confusion. "Notably, the actual iterative expansion ratio is 1.5 in the x, y and z planes even though we applied a 4x protocol. This outcome is likely due to the sample being too large for homogeneous gel polymerization."

2. On line 23 of page 8, it is mentioned that long-term exposure to mounting adhesive causes a smaller expansion factor. Does it cause non-homogeneous expansion across z-direction?

Reply: Thanks for raising the anisotropic issue; from imaging data, we did not observe z-direction non-homogeneous expansion compared to xy directions.

3. Is there any specific reason for setting the digestion temperature at room temperature instead of at 37 degree? Can the authors demonstrate that the digestion is complete and the tissue is cleared?

Reply: Thanks for the question. We compared sample size and fluorescence signal intensity after digestion at 37C and room temperature and found that the sample we tested (fly brain, cells, spheroids, up to 100um tissue slices) could be digested completely at room temperature, and fluorescence signals were maintained better. Also, digestion of the *Drosophila* brain at room temperature has been previously demonstrated (Lillvis et al., 2022 (eLife)).

4. Is there any specific reason for using ddH2O instead of salt solution (or buffer) to dissolve and mix monomers?

Reply: Thanks for the question, no NaCl was added to KA or PKA monomer cocktail as the salt did not help in terms of gel stability and expansion ratios, the simplified version of KA (KOH+Acrylic Acid+MBA) and PKA (DMAA+PA) formula was invented to avoid using complicated chemistry in expansion protocol, and the resulting gel performance fully satisfied our purpose. Our previous work TT-ExM (Wang, U.-T. et al. "Protein and lipid expansion microscopy with trypsin and tyramide signal amplification for 3D imaging", *Scientific Reports*, 13, 21922 (2023)) also demonstrated using a low salt buffer in the digestion step helped increase expansion ratio.

5. Is there any specific reason for not degrading the template gel used for re-embedding for the *re*-KA-ExM and *re*-PKA-ExM?

Reply: Thanks for the question. We found that the NaOH disruption step was too harsh on fluorescence signals, and by eliminating this step, the second gel expanded to the same size.

6. To make it easy for the general readers of Nature Communications the authors might consider dividing the result section into shorter subsections with descriptive titles.

Reply: Thanks for the helpful suggestion. We added the descriptive titles for each subsection, as shown in the revised manuscript.

Comments on figure:

1. For supplementary figure 1g, it is not clear how the experiments and statistics were done. Is it 9 identical cells expanded in one KA-ExM gel? The number of technical replicates is not stated.

Reply: Thanks for the reviewer pointing this out. We have added the descriptions about how these 9 identical cells were chosen in **Supplementary Figure 1**; note that three coverslips hosting cultured cells were analyzed separately, with three cells chosen from each coverslip for pre- and post-expansion size measurements. There is one coverslip, for example, shown below. We took the pre-expansion confocal images for the cultured cells and performed the KA-ExM. The KA-ExM gel was returned to the confocal microscope for the fluorescence images. By the imaging analysis, we found the same areas shown in the white box and found the cell nucleus for the expansion ratio measurement marked in the circles.

2. Please make sure that the structure of the DMAA/PKA polymer chain in Figure 3a is correct. Will there be repeated KA groups?

Reply: Thanks to the reviewer for pointing out this typo. We are sorry for the wrong chemical formula. We have revised the new structures of the DMAA/PKA polymer chain in **Figure 3a**.

3. In supplementary figure 4, it seems like the stitching isn't great. Is it possible to replace them with better images? Also, the gel doesn't seem to have isotropic expansion.

Reply: Thanks for the reviewer's suggestion. We have tried our best to redo these stitching better as shown in the new **Supplementary Figure 5**. Since these are epi-fluorescence images for pre-

and post-PKA-ExM on *Drosophila* brain for 7 trials shown in (a), the non-uniform intensity and some angle distortion due to very thick hydrogels and the sample positioned not right to the image plane. Here we would like to give a roughly physical size of the whole fly brain after iterative ExM. According to the new **Supplementary Table 4**, there is some x/y ratio that behaves not isotropic.

4. In figure 4 b&c, it seems like the gel expands more in the direction of the length of the rectangle. The relative location of the orange nuclei is not consistent between b and c. Can the figure legend be re-written? It's not clear how b and c are related to panel a.

Reply: Thanks for the reviewer's concern about the image orientation. Because the expanded sample was conducted using iterative imaging following the pipeline shown in **Supplementary Figure 7a**. Following PKA ExM, the hydrogels are mounted and transferred to the Δ BLX chamber for 3D whole-brain imaging. After lightsheet imaging, the hydrogel is removed from the water chamber and subjected to *re*-PKA ExM. This huge *re*-PKA fly brain (>2 cm, shown in **Supplementary Figure 5b**) was cut in half and placed back into the Δ BLX chamber and underwent lightsheet whole half-brain imaging for characterization with different orientation (very hard to make the exact mounting position as the PKA sample by tweezers). When we try to register these two 3D lightsheet images (PKA vs *re*-PKA), we have to match them in 3D image space. And here comes the problem inherent in the imaging objective (NA=0.6): the axial resolution is worse than the lateral resolution. Therefore, the worse resolution in the z direction happens in the differently oriented PKA and *re*-PKA samples. The registration shows the anisotropic between these two identical data sets of cells. We found there is angle discrepancy about 15 degree with respect to the imaging lens (as shown below). The overlapped slice view of PKA-ExM (12x) and *re*-PKA ExM (32x) for the same selected area in the optical lobe, where the PKA ExM grows image multiplied the scale factors of 2.9, 3.0, and 4.1 separately in x,y,z directions to match the *re*-PKA-ExM grey raw image in 3D with 15-degree rotation. 3D renderings of the same selected regions in the grey area, representing PKA-ExM (left) and *re*-PKA-ExM (right). We redo **Figure 4** and a new **Supplementary Figure 8** to make it easier to understand for the iterative imaging. We focus on the area marked by the yellow box to compare PKA vs *re*-PKA at the single-cell level from the expanded fly brain data.

5. Is referencing Supplementary Fig. 1c in Page 6 line 2 correct?

Reply: Thanks to the reviewer for pointing out this typo. We have corrected it to **Supplementary Fig. 1d** on page 6.

Reviewer #3 (Remarks to the Author):

An important challenge in understanding how neurons act within circuits is resolving the components of neurites and synapses. Although this has been previously hampered by light microscopy's diffraction limit, expansion microscopy (ExM) has allowed scientists to resolve small structures with impressive resolution. Most ExM protocols involve a sodium acrylate-based hydrogel, which, although effective, can be watery and unstable, limiting the degree of expansion and the shape integrity of the samples. Tian, Lin et al., solve this issue by making two new types of hydrogel—one using potassium hydroxide and acrylic acid, and the other using commercially available PKA. The new gels have increased mechanical stability—the KA-based ExM can be sectioned using a vibratome—and improved expansion ability—the PKA-based ExM can induce a 32-fold isotopic expansion.

Imaging an expanded sample can also provide distinct challenges, most notably, a decrease in fluorescence intensity and a reduced Z-range due to the limited operational distances of the objectives. Tian, Lin et al build a Bessel beam light sheet to solve this issue. With two longer working-distance objectives, they are able to image expanded samples with less difficulty.

The group then applies KA/PKA-based ExM to the delicate neurites of the *Drosophila* optic lobe, imaging dopaminergic neurons and optic lobe neurons with nanometer resolution. The images are impressive and they are able to visualize gaps in the neurites that they believe correspond to mitochondria, and densities that correspond to synapses. This protocol is an exciting addition to the ExM repertoire, and will prove quite useful to many scientists, whether studying *Drosophila* or another model organism.

Although the images are quite striking and beautiful, the figures are a little bit crowded, and could use some rearranging to make the paper easier to read. Also, while the paper is experimentally sound, there are a few experiments that should be performed to provide more evidence for predictions made within the manuscript. That said, any limitations I found were rather minor, and I believe that the paper should be accepted with minor revisions.

Reply: We appreciate the reviewer's summary of our work, including no sodium acrylate, good mechanical strength for vibratome sectioning, and greater expansion ratios. Moreover, the reviewer highlighted the present Bessel lightsheet microscope enabling the largely expanded samples with two longer working-distance objectives on the various biological samples to visualize the detailed structures. And thank the comments and suggestions for the figures and experiments. We have added new figures and new Supplementary figures. Hopefully, with these revisions, the manuscript could be published in the journal for a broader readership.

Typos/text edits:

- TH-Gal4 (tyrosine hydroxylase) is not defined in the manuscript. Please add a sentence to suggest that you are labeling dopaminergic neurons in the main text.

Reply: Thanks for the reviewer's suggestion. We have added the following sentence about TH-Gal4 labelled neurons, and cited the related reference. “The rate-determining reaction in the biosynthetic pathway of *Drosophila melanogaster* that produces the neurotransmitter dopamine is mediated by tyrosine hydroxylase (TH).²⁴” in page 6.

- Line 12: There is an extra period in the sentence “determined the 3D expression pattern of transgene, TH-GAL4 in expanded fly brain..”

Reply: Thanks for the reviewer's correction; we have removed the extra period.

- Page 9, Line 18: “expaned samples” should be “expanded”

Reply: Thanks for the reviewer's correction, we have corrected the spelling.

- Page 12, line 12-13: Change “where one segmented lamina monopolar neuron L3 (membrane GFP, shown in blue) among photoreceptors in red.” to “shown in blue) IS among photoreceptors in red.”

Reply: Thanks for the reviewer's careful check; we have corrected this and rewrote the sentence as follows, “we show a representative multi-color volumetric image of one segmented lamina monopolar neuron L3 (membrane labeled with GFP, shown in blue) among photoreceptors (in red)” in page 12.

- Page 12: line 21: Change “involvelemt” to “involvement”.

Reply: Sorry for the typo. We have corrected this and rewrote the sentence: “According to EM data from NeuPrint (Supplementary Fig. 11g, 11h, and 11i), the Dm9 neuron presumably fills up this space during development.”

- Figure 6: change “gree” to “green.” (line 5, page 29)

Reply: Sorry for the typo. We have corrected this.

- Supplementary Figure 10: DAPI misspelled as “DPAI” (line 3)

Reply: Sorry for the typo. We have corrected this. “The 1 mm thick side view of the three-color optical lobe image with nuclei (DAPI, blue), sparse labelling visual neurons (mCD8GFP, green) and photoreceptors R7 and R8 (GMR-RFP, glow).”

- Page 13, line 16: change “routin” to “routine”

Reply: Sorry for the typo, we have corrected this. “Considering the dilution effect on the fluorophore intensity, stability of the chemical reagents and sample handling of the expanded tissues, in situ KA-ExM is preferable for most daily routine experiments.”

- Pag 13, line 17: change “fluorescen” to “fluorescent”, “singals” to “signals”

Reply: Sorry for the typo; we have corrected it. “Given that spatial resolution is often critical, optimizing PKA-ExM could be the best option for increasing fluorescent signals, especially by altering experimental conditions such as the imaging buffer.” In page 13.

Figure edits:

- It would be nice to have a better definition of “expansion factor”: i.e. Yellow scale indicates the actual size of the image divided by the expansion factor.

Reply: Thanks for the suggestion. We have added this description to the main text on page 8 “Note that the white scale bars for all panels are in real physical units while the yellow scale bars indicate the actual size of the image divided by the expansion factor.”

- Most of the figures are crowded and have too many overlapping components. Some figures would benefit from being cleaned up and put into insets:

Reply: Thanks for the reviewer’s suggestion. We have restructured some figures, especially for **Figure 4**, and tried to make more supplementary figures to make it easier for the readers to understand the side-by-side comparison. Sorry for the crowded and busy figures with detailed structural comparisons in order to make the manuscript solid and sound.

-Figure 1F: Put the smaller ellipsoid body in a box at the corner of the figure, not within the ellipsoid body itself.

Reply: Thanks for the reviewer’s suggestion. We have revised the figures accordingly, and put the original ellipsoid body in a yellow dashed box at the corner of the figure1f.

-Figure 3B: Put the 1x brain into a separate inset.

Reply: Thanks for the reviewer’s suggestion. We have revised the figures accordingly, and put the original 1x fly brain image in a red dashed box as an inset in Figure 3b.

-Figure 4: it is too challenging to tell the difference between the samples with the top right overlaid on top of the PKA-ExM. Would be better as an inset in the left corner.

Reply: Thanks for the reviewer's suggestion. We have revised Figure 4 as new **Figure 4**, where we removed all the overlapping components in Figure 4a, and only show the selected area for the PKA- and *re*-PKA ExM comparisons in **Figure 4a** marked with a yellow box. **Figures 4b** and **4c** show the zoomed area for a slice view for the area marked in **Fig. 4a** for a 12x expanded fly brain. **Fig. 4c** is the enlarged views in the dashed box in (b) of the same nuclei colored red and green for PKA-ExM (12x, inset) and *re*-PKA-ExM (32x, main image), with the same scale bar for the inset and main image. **Fig. 4d** only shows the 3D volume rendering of six nuclei for iterative expansion. Additionally, we have added new **Supplementary Figure 8d~8g** for the detailed comparisons. Hopefully, with these changes, the figure becomes easier to read.

- Please make sure that all the components of each figure are defined:

-Supplementary figure 4: There is too little description within the figure legend: Please mention that the small insets are pre-expansion samples, and the larger images are post-expansion.

Reply: Thanks for the reviewer's suggestion. We have added the required information in Supplementary figure 4, now in revised **Supplementary Figure 5**, "The epi-fluorescence imaging for pre- and post-PKA-ExM on *Drosophila* brain for 7 trials shown in (a) where the original fly brains shown in grey color and dashed box and expanded fly brain shown in the glow color."

-Figure 2A-C Legend: please write out that the green box in 2D is an inset of the dotted box at Figure 2A (also, the proportions of the two green rectangles do not seem to be the same size).

Also, please define the yellow/blue boxes in Figures 2B/C in the legend as well.

-Figure 2G-J: define what the white/blue dotted circles are in your figures.

Reply: Thanks for the reviewer's suggestions. We modified the legend of Figure 2 as, "(a) Confocal image of dopaminergic neurons where PRW neuronpil marked in a green box in the original (1x) TH-GAL4, 20XUAS-6XGFP/+ *Drosophila* using a NA=1.1 objective. (b) and (c) Two-color images (purple, nuclei; heatmap, dopaminergic neurons) of the identical TH-GAL4 fly brain stained with DAPI shown in (a) after in situ KA-ExM (8x) and re-KA ExM (13x). The identical PRW neuronpil highlighted in the yellow and blue dashed boxes for iterative expansion process. (d) A green box for confocal image of the partial area within the green dashed box in (a) and enlarged view of the punctate structures outlined by the red dashed box. (e) and (f) Enlarged views of the regions highlighted by the colored dashed boxes in (b) and (c), respectively, with the same scale bar for the same neuropil following in situ KA-ExM (8x) and re-KA ExM (13x). (g) and (h) Zoomed-in views of the selected areas within the white dashed boxes of (e) and (f). The marked areas (white dotted circles) show the same biological structures after 8x and 13x expansion. (i) and (j) Raw images of the same punctuate structures (highlighted in blue dotted circels) after in situ KA-ExM and re-KA-ExM. Note that the scale bars for all panels in real physical units (white font) vary according to the expansion factor (yellow font)."

- Supplementary figure S6: it would be nice to have a zoomed-in inset where you can better see the colocalization of the KA-ExM with the re-KA-ExM. Also, what is the reasoning why the brain halves are labeled in two different colors (cyan/magenta)?

Reply: Many thanks to the reviewer for pointing this out. Supplementary Figure S6 (now **Supplementary Figure 8**) is about PKA and *re*-PKA ExM. The zoomed-in view for the detailed overlapping is demonstrated in the main text for **Figure 4**, where we show the cellular arrangement between PKA and *re*-PKA ExM in **Figures 4b** and **4c**. The reason we show the *re*-PKA ExM fly brain in two colors is that we physically cut this more than 2-cm sized gel into half (one shown in cyan and the other one shown in magenta, the magenta one is the one shown in **Figure 4** in the main text) in order to fit into our imaging chamber for lightsheeting imaging for the measurement of expansion ratios so that we could transform these different sets of data into one coordinate system to fit the PKA-ExM whole fly brain shown in glow color.

- Supplementary Figure 10 is cited before Supplementary Figure 9. Please switch the order of the labeling.

Reply: We have checked the ordering. In the revised manuscript, they are now **Supplementary Figure 11** and **Figure 13**, respectively.

Experimental edits:

- Supplementary figure S7: Would be nice to have a corresponding NCadherin stain to show medulla layers for reference to the reader.

Reply: Thank you for the suggestion. We agree that including a landmark stain would improve the clarity of the figure. In response, we have added a schematic illustration in the new **Supplementary Figure 9a** left panel and a whole-mount image (right panel) of the optic lobe. In the new images, photoreceptor R7 and R8 axons, visualized using the 24B10 antibody, serve as landmarks for the M6 and M3 layers, respectively, to help orient the reader. The following information is added in the new **Supplementary Figure 9**. “The confocal images of Tm5a neuron in the optical lobe acquired in Zeiss LSM 880 with 20x NA=1.1 objective lens. (a) Schematic illustration of the retina, lamina, and medulla structures in the Drosophila optic lobe. The transmedullary neuron Tm5a (blue) extends its dendrites predominantly into the M3 layer (R8 axon terminal, cyan), M6 layer (R7 axon terminal, magenta), and M8 layer. Tm5a neurons were analyzed in flies expressing Tm5a split-LexA > CD4tdGFP. The GFP signal (green) was detected using an anti-GFP antibody. Photoreceptors R7 and R8 were visualized using 24B10 antibody (magenta). Scale bar: 20 μ m.”

- The authors suggest that the knots in the synaptic areas are filled by mitochondria. This would be very simple to test with a mitotracker stain + GFP colabel.

Reply: Thanks for the reviewer pointing out this. To validate the speculated mitochondria structures based on the optical images in **Figure 2j**, **3d**, and **Supplementary Fig. 6c**, we generated a fly strain, UAS-mCherry.mito.OMM/+; TH-GAL4, 20XUAS-6XGFP/+ fly to specifically label the mitochondria and performed two-color PKA-ExM lightsheet imaging. The related data is present in the new **Figure 7a**, shown below to the right, and **Supplementary Figure 14a**, shown below, which supports our finding that the empty space is occupied by mitochondria. Two-color MIP image of GFP-labeled TH-GAL4 (green) and mCherry-labeled mitochondria (red) in the PRW region of the PKA-ExM fly brain.

- Figure 5D: The reconstructed image doesn't give enough context. Please provide the original fluorescence image used to separate the neurites.

Reply: Thanks for the reviewer's suggestion. We have added the following raw images for the series of z slices of the Tm5a neurites at z interval of 6 μ m. The red dashed box shows the area where highlighted in **Figure 5D**. These new raw images are supported as new **Supplementary Figure 10**.

- Supplemental Figure 8: the authors argue that ExM provides a more impressive image quality than EM. Would be nice to have some sort of quantification that shows this, rather than an “improved” tracing. It would also be nice to have a table that compares SA-ExM to KA-ExM.

Reply: Thanks for the reviewer's suggestion. We have added the new **Supplementary Table 1** that includes our KA- based ExM with other SA-based ExM. For the EM part, we have to tone down the argument about “improved”, since all the imaging techniques have their own pros and cons. On page 12 “Indeed, the optical images presented in Fig. 5d reveal comparable profiles as in the EM-imaged Tm5a neuron shown in Supplementary Fig. 11a.” For example, it is not fair to make the judgement on this single example from EM, not representative for all Tm5a neurons, individual diversity should be considered. In order to highlight the benefit of fluorescence KA-ExM with a greater expansion ratio, we added the segmentation data about densely labelled Tm5a neurons in the revised **Figure 5b**, new **Supplementary Figure 12** and **Supplementary Movie 5**. We do observe the diversity of the dendritic processes (some in complicated while some in simple profiles) as shown in the skeletonized 15 Tm5a neurons in the *re*-PKA ExM fly optical lobe. For

fluorescence-based ExM, it is better to conduct a specific and comparative systematic analysis. The top view (a) and side view (b) of a group of skeletonized Tm5a neurons in the M3, M6, and M8 layers are shown below. (c) The top view (left) and side view (right) of the individual and colored Tm5a neurons with dendritic processes in M6 layer marked with white arrowhead in (b).

Manuscript ID: NCOMMS-24-32341A

Paper title: Rapid lightsheet fluorescence imaging of whole *Drosophila* brains at nanoscale resolution by potassium acrylate-based expansion microscopy

Authors: Tian, X. and Lin, T.-Y. *et al.*

Reply point-by-point

Reviewer #1 (Remarks to the Author):

Thanks to the authors for addressing my comments. I have a few additional comments/suggestions based on the revisions.

1) If I understand the effective resolution correctly, the effective axial resolution achieved at ~40x is ~75nm. State of the art segmentation approaches used in EM would indicate that this is not sufficient resolution for densely labeled neuron segmentation and I would say this is not exactly a resolution that is similar to EM. Therefore, I would suggest the authors change the language in the abstract and conclusion of the manuscript that suggests the effective resolution is comparable to EM and will allow dense neuron segmentation. There are many positive things to say about the approach, but I don't think those conclusions are supported by the current methods advances.

Reply: Thanks for the reviewer's suggestion. We totally agree with this suggestion. And the related descriptions already shown in the abstract " Page 3 Line 12 " we image the centimeter-sized fly brain at an effective resolution comparable to electron microscopy," and in the conclusion part, Page19 Line 3, we add "By means of a greater sample expansion ratio from iterative process and axicon-based Bessel lightsheet to gain effective spatial resolution, imaging the nanoscopic molecular architecture of features such as the synaptic active zone, T-bar shown in the EM...." We will keep working on this research direction to explore more interesting research. Thanks for the reviewer's comment and suggestion. We will keep this in mind. Truly, different imaging modalities have their pros and cons and we have to understand and use them correctly.

2) Thanks for the new documentation of expansion factors in Sup. Figs 3-5 and clarity on this in the manuscript. Given the manuscript is heavily about *Drosophila* brain PKA-ExM and rePKA-ExM, the images shown in the rebuttal but not the manuscript seem helpful. Do data from these images make up part of numbers in Sup. Table 5? I don't think it's necessary to include the images from the rebuttal in the paper, but please add N to Sup. Table 5 to indicate the number of samples that went into these numbers. It would also be better to provide an actual average +/- SD instead of approximate given you have the data.

Reply: Thanks for the reviewer's suggestion. We add the N to the Sup Figure 5, "Expansion ratio in long axis of fly brain is 14.2 ± 0.98 ; expansion ratio in short axis is 15.4 ± 1.91 based on 7 measurements. 2-dimensional distortion in (a) was calculated by comparing the size of the same fly brain along two directions before and after expansion resulted in 0.93 ± 0.11 ."

a. On this point, the authors state that it is difficult to obtain expansion ratios in very large samples because it requires high-res imaging to get these numbers. However, it should be possible to calculate expansion factors from the widefield images of the brains pre and post expansion. Or at least post expansion in relation to the average size of the *Drosophila* brain. It looks like these images exist for the samples discussed. If that's not the case, I wouldn't suggest new experiments to get this, but if you have the data I would use it to calculate expansion factor using a common metric instead of whole brain size in some instances, central complex size in others, axon diameter in others, etc.

Reply: Thanks for the reviewer's suggestion. We have tried a lot for such kind of measurements but epi-fluorescence fails to image largely 40x re-PKA expanded fly brains at the size of 2 cm long and >1 cm thick (included surrounded gel). In addition, due to ~64000 dilution of fluorophores, the fluoresce signals becomes undetectable by wide-field illumination fluoresce microscope. Here we present two photographs shown below, the left one is 10x fly brain detectable by epi-fluorescence (~ 5 mm fly brain could be observed); the right one is the re-PKA ExM fly brain with >40 expansion ratio becomes undetectable even though with high sensitivity camera. Therefore, we construct a lightsheet microscope in order to capture the whole expanded fly brain. Last but not least, we are working on building specialized epi-fluoresce microscope to do the research that the reviewer mentioned. Hopefully, in the near future we could share our experiences on this development.

3) Line 22, page 9 indicates blue arrow figure 3c indicates empty spaces in the neural fibers, but it's very difficult to see this in the figure. Further, I would suggest stating what the arrows are meant to indicate in the figure legends – not just in the main text.

Reply: Thanks for the reviewer pointing this out, probably because of 3D rendering, it is hard to see the structures inside, we add an enlarged slice view for this area in the revised Figure 3c. And we also add the description about the markers in the figure legend. At Page 36 Line 8

“(c) The tile demarcated by a white dashed box in (b) rendered in 3D side view to demonstrate axial resolution. The two sub-volumes highlighted by green and blue dashed cubes are shown at

right, revealing punctate (white arrow) and fibrous neuronal structures (yellow and blue arrows). A 2x enlarged slice view marked with blue arrow shown in the inset.”

4) I would suggest moving the new data introduced in the discussion to the main body of the manuscript.

Reply: Thanks for the reviewer’s helpful suggestion, we have moved the **Sup Figure 12** to the main body of the manuscript as new **Figure 6**, at Page 14 Line 7, “Moreover, a group of densely packed Tm5a neurons could be segmented in a largely expanded fly brain as displayed in the white dashed box in **Fig. 5b**, where ~15 neurons are colored and skeletonized as shown in **Fig. 6**. The top and side view of the skeletonized expanded Tm5a neurons with colors in **Fig. 6a** and **6b**, respectively. The adjacent Tm5a neurons spanning lateral branches extend to reach neighboring columns in M3, M6, and M8 layers. **Figure 6c** shows top and side views of the representing segments for the dendrite field in M6 layer (white arrow in **Fig. 6b**). Supplementary Fig. 12 and Supplementary Movie 5 demonstrating the complete segmentation and various dendritic processes in M6 layer.” And at Page 42 Line 1, “**Fig. 6**. 3D skeletonization feature of densely packed Tm5a neurons in the re-PKA ExM fly optical lobe with 40 plus expansion ratio. (a) Top view of a group of skeletonized visual neurons Tm5a based on the segmentation data in the white dashed box of Figure 5b. (b) Side view of these transmedullary neurons in (a) with their dendrites extending into the M3, M6, and M8 layers from top to bottom. (c) The top view (left) and side view (right) of the individual and colored Tm5a neurons with dendritic processes in M6 layer marked with white arrowhead in (b). Note that all the images sharing the same scale bar.”

Moreover, we also moved the Supplementary Figure of re-KA and re-PKA ExM based on HCT spheroid to the main figures, as **Figure 9**. At Page 16 Line 11 “For sample diversity, the validation of re-KA and re-PKA ExM on the spheroids generated from the HCT 116 cell line was performed. In Supplementary Fig. 15a and 15b, we present the expansion ratio of ~100 μm spheroids before and after expansion, resulting in ~20x for re-KA and ~40x for re-PKA, respectively. Figure 9a shows the image of original spheroid and Fig. 9b displays the expanded spheroid has ~4 mm size with maintained morphology after re-PKA ExM with its original size shown in the inset at the same scale. An enlarged 3D rendering and slice view for a mitotic cell (>300 μm) inside the spheroid shown in Fig. 9c and 9d with detailed chromosomal structures or an interphase one in Supplementary Fig. 15c and 15d.”

5) I think it is the case that Gao et al., 2019 used a 4X, not an 8X, protocol and that Lillvis et al., 2022 used an iterative 8x protocol. This differs from what is indicated in Sup. Table 1.

Reply: Thanks for the reviewer pointing this out. We corrected these two typos in the revised Sup Table 1.

Reviewer #2 (Remarks to the Author):

The authors have satisfactorily addressed all of my prior concerns.

Reply: Thanks for all the comments from the reviewer.

Reviewer #3 (Remarks to the Author):

In this paper, Tian, Lin et al., improve the resolution of traditional expansion microscopy by making two types of hydrogels with increased mechanical stability to improve resolution of imaging samples such as whole-mount *Drosophila* brains. The gels produced by the authors not only offer improved resolution, but they also simplify the protocol for the end user. Although a few controls were missing from the initial draft, the authors have performed several experiments and provided adjustments to the paper that improve its quality. The use of spheroids has added rigor to the quantification of the expansion factor, and the use of Mitotracker more conclusively shows what the authors purported in the initial draft. Finally, the construction of a Supplemental Table comparing previous expansion microscopy systems to their own better shows how their work is an advancement compared to previous protocols in the field. I therefore suggest that the paper be accepted.

One small correction however: I spotted another typo in the figure legends (page 25, line 15: "circels" should be "circles")

Reply: Thanks for all the input from the reviewer and pointing the typo out. We corrected it in the final revision.